# Scalable semitransparent organic solar cells with robust film thickness tolerance for building-integrated photovoltaics

Tong Wang[1,6], Jin Fang[2,6], Hao Zhang [1], Chenyang Tian [1], Yuhan Wang[1,3], Zhen Fu[4], Wenjun Zou[2], Dan Deng [1], Xiaotao Hao [4], Chang He[5], Jianqi Zhang [1] ✉ & Zhixiang Wei [1,3] ✉

Building-integrated photovoltaics (BIPVs) is a promising application for semitransparent organic solar cells (ST-OSCs). However, conventional ultra-thin (<80 nm) active layers for ST-OSCs, while balancing transmittance and efficiency, limit the cell-to-module efficiency remaining ratio (CTM) below 56%. Here, we achieve high semitransparency and efficiency in ST-OSCs with reasonable active layer thickness by manipulating the aggregation of acceptors in various donor-diluted blends processed with non-halogen solvent in ambient air. Using PM6:Qx-p-4Cl as a model system, we elucidate a unique film-formation mechanism and charge generation process, demonstrating that the fiber network and suitable aggregation size are crucial for ensuring higher performance in donor-diluted ST-OSCs. The 1 cm² donor-diluted ST-OSCs with active layer thicknesses of 119 and 301 nm exhibit high light utilization efficiencies (LUEs) of 4.04% and 3.02%, respectively. Notably, a 100 cm² module demonstrates a CTM ratio of ~85% and a LUE of 3.32%, owing to its high film thickness tolerance, setting a new benchmark for large-area semitransparent modules. Furthermore, we demonstrate the feasibility of BIPVs in terms of power generation, energy storage, and temperature control through a scale-down model with a 600 cm² power-generating window. These results reveal promising prospects for ST-OSCs in real-world applications.

Building-integrated photovoltaics (BIPVs) are pivotal for revolutionizing sustainable urban architecture, with semitransparent photovoltaics emerging as an essential technology to realize this vision through transforming conventional building envelopes from passive energy consumers into self-powered structures. Organic solar cells (OSCs) emerge as promising candidates for semitransparent photovoltaics due to their unique benefits of optical bandgap regulation and the intrinsic semitransparency of the acceptor, thus demonstrating considerable potential for BIPVs.[1–7] The development of BIPVs focuses on semitransparent properties, device efficiency, and large-area module processing technology for semitransparent organic solar cells (ST-OSCs).[8,9] Specifically, the key driver in the preparation of high-performance BIPVs is realizing an optimal balance between average visible transmittance (AVT) and power conversion efficiency (PCE) in large-area semitransparent modules (ST-modules).

Compared to opaque OSCs, the performance of ST-OSCs is commonly evaluated using light utilization efficiency (LUE), which reflects the trade-off between AVT and PCE (LUE = PCE×AVT).[10–12] A

[1]CAS Key Laboratory of Nanosystem and Hierarchical Fabrication, National Center for Nanoscience and Technology, Beijing, P. R. China. [2]Hyper PV Technology Company Limited, Jiaxing, P. R. China. [3]University of Chinese Academy of Sciences, Beijing, P. R. China. [4]School of Physics, State Key Laboratory of Crystal Materials, Shandong University, Jinan, Shandong, P. R. China. [5]Institute of Chemistry, Chinese Academy of Sciences, Beijing, P. R. China. [6]These authors contributed equally: Tong Wang, Jin Fang. ✉e-mail: zhangjq@nanoctr.cn; weizx@nanoctr.cn

comprehensive analysis of reported data reveals two primary approaches to enhancing LUE. The first strategy involves optimizing film thickness, where researchers have systematically demonstrated that preparing an ultra-thin active layer (<80 nm) can overcome the LUE limitations for ST-OSCs.[8,12–20] The second strategy focuses on further modulating the donor-to-acceptor ratio (D:A ratio) in thin films. Given that the visible absorption in high-performance organic photovoltaic systems predominantly originates from donor materials, diluting donor content effectively enhances the AVT but significantly reduces PCE.[14,21] Notably, both strategies maintain the optimal LUE threshold at approximately 60–80 nm film thickness by utilizing harmful halogenated solvents (e.g., chloroform and chlorobenzene) in nitrogen-filled glove box. Such ultra-thin active layer faces three critical obstacles in large-scale manufacturing: (1) exacerbated performance degradation when scaling up production, (2) significant biotoxicity of halogenated solvents and (3) critically low production yields resulting from stringent thickness control demands, which were not taken into full consideration in previous reports.

To quantitatively evaluate the performance loss during upscaling for OSCs, the cell-to-module efficiency remaining ratio (CTM) is proposed, which is calculated from the ratio of performance between the monolithic device and the corresponding large-area module.[22,23] Impressively, the CTM of ST-OSCs is generally lower than that of opaque OSCs. Especially for the ST-module with an area above 100 cm², the reported CTM of the 100 cm² ST-module is around 56%, which lags far behind the 100 cm² opaque module reported in previous works (~ 88%).[23,24] In addition to inherent electric and geometric losses of the module design, this performance gap primarily stems from the increased active layer thickness during scaling up production, which reflects the fundamental challenge of fabricating very thin film using scalable fabrication technologies. However, the large active layer thickness leads to enhanced recombination, poor charge extraction, and reduced transmittance, resulting in exacerbated degradation of LUE for ST-OSCs.[25,26] Consequently, it is necessary to optimize the AVT of the thick-film active layer and improve the film thickness tolerance for preparing high-performance upscaling ST-modules.[27] In principle, it is recognized that the donor-dilution strategy is an efficient strategy for ensuring high transmittance in the thick active layer.[5,17,28] However, the unsatisfactory D:A ratio remains challenges of the excessive aggregation and disordered structure for the non-fullerene acceptor, leading to imbalanced charge transport and competition between charge extraction and recombination.[24,29,30] Recently, our group found that the optimal D:A ratio of the slot-die-coated device (1:1) shows a higher acceptor proportion than that of the spin-coated device (2:1) in some systems.[31] This finding inspires us to consider that the combination of donor-dilution strategy and slot-die coating would represent a promising approach for improving CTM during large-scale production of ST-OSCs and provide new access to realize high-performance BIPVs.

Herein, we achieved high-performance thick-film slot-die-coated ST-OSCs by systematically manipulating the morphology of donor-diluted film (D:A ratio = 1:3) processed with non-halogen solvent in ambient air, and then obtained an excellent CTM in ST-modules for the demonstrated application under outdoor conditions. First, a high LUE of 4.04 % with PCE of 10.57% and AVT of 38.21% are obtained in the 1 cm² slot-die-coated semitransparent device (ST-device, D:A ratio of 1:3) with an active layer thickness of 119 nm. Even when the thickness increases to 301 nm, which is three times more than that of previously reported results,[8,12–20] the LUE remains high at 3.02%, demonstrating exceptional film thickness tolerance. Next, by tracking acceptor aggregation and excited state behavior, we elucidate the unique aggregation mechanism and charge generation process in donor-diluted slot-die-coated bulk heterojunction (BHJ). Finally, the 100 cm² printed ST-module prepared by the donor-dilution strategy yields a LUE of 3.32% and a CTM of around 85%. When transitioning from the

laboratory to the outdoor environment, we utilized a scale-down house model with a 600 cm² power-generating window (six modules in parallel) to demonstrate the feasibility of BIPVs in terms of power generation, energy storage and temperature control. Overall, our work elevates the performance of upscaling ST-OSCs to a higher level and highlights the potential of donor-diluted ST-devices for real-world applications.

## Results
### Photovoltaic properties

The chemical structures of the PM6:Qx-p-4Cl system and thickness-normalized absorption spectra of the blends with different D:A ratios are illustrated in Fig. 1a, b. To investigate the photovoltaic properties of spin-coated and slot-die-coated devices with different D:A ratios processed with non-halogen solvent in ambient air, the PCE versus D:A ratios of large-area opaque devices (1 cm²) based on PM6:Qx-p-4Cl is presented in Fig. 1c and Supplementary Fig. 1. And the corresponding EQE spectra are shown in Supplementary Fig. 6. The optimal D:A ratio of the spin-coated device is centered at 1:1.5 (Supplementary Table 2). The corresponding device shows PCE of 13.80%, along with open-circuit voltage ($V_{OC}$) of 0.89 V, short-circuit current density ($J_{SC}$) of 21.29 mA cm⁻² and FF of 72.61%. For the devices fabricated by slot-die coating with 90°C coating substrate, the large-area device based on PM6:Qx-p-4Cl exhibits the highest PCE of 12.79% at D: A ratio of 1:3, indicating that the slot-die coating method can realize better performance in donor-diluted devices. Besides, we summarized the photovoltaic parameters of PM6:Y6, PM6:L8-BO, and PM6:Qx-1 with different D:A ratios in Supplementary Fig. 2–9 and Supplementary Table 3–5. All the slot-die-coated systems show high performance at a high acceptor ratio. In particular, the PM6:Y6 devices achieve the optimal efficiency when the acceptor ratio exceeds twice that of the donor. However, the best performance of PM6:L8-BO and PM6:Qx-1 devices is obtained at D:A ratios of 1:1.7 and 1:1.5, respectively. Further dilution of donor content leads to a decline in device performance.

Benefiting from the lower donor content at optimal D:A ratio of PM6:Qx-p-4Cl, we further explore the photovoltaic properties of ST-OSCs. A 20 nm ultra-thin Ag electrode is utilized to fabricate the slot-die-coated large-area ST-devices. As seen in Fig. 1b–d and Supplementary Fig. 11, diluting donor content within the active layer significantly reduces the absorption in the visible range, realizing high AVT in donor-diluted ST-OSCs. The highest PCE is obtained at D:A ratio of 1:3 with AVT of 15.43%, which is consistent with the corresponding opaque devices (Supplementary Table 6-7). As shown in Fig. 1e, Supplementary Fig. 13, and Supplementary Table 8, we compared the AVT of 1:1.5 and 1:3 active layers with different thicknesses. The AVT of donor-diluted blends (D:A = 1:3) is significantly higher than controls (D:A = 1:1.5) at the same film thickness. More importantly, the donor-diluted blend with a large thickness (205 nm) exhibits a high AVT of 54.68%. However, achieving an AVT above 50% requires a thickness below 100 nm for the 1:1.5 blend. With increasing film thickness, the advantage in transmittance of donor-diluted blends becomes more pronounced. The parameter Δd quantifies the thickness difference between 1:1.5 and 1:3 blends under constant AVT. At AVT of around 70%, Δd is around 80 nm. For thicker blends, Δd is increased to above 125 nm at AVT around 55%. These results suggest that the integration of donor dilution and slot-die coating is a viable method for the mass production of high-performance ST-OSCs, exhibiting high AVT in thick films.

To further improve the photovoltaic and semitransparent properties, we adopted a conventional device structure, and a 35 nm MoO₃ anti-reflective coating (ARC) layer was coated on an ultra-thin Ag electrode.[2,8,18,32,33] The current density (J)-voltage (V) curves and detailed parameters of ST-OSCs with ARC layer are presented in Fig. 1f, Supplementary Fig. 1 and Supplementary Table 6. The corresponding EQE spectra are shown in Supplementary Fig. 10. The slot-die-coated

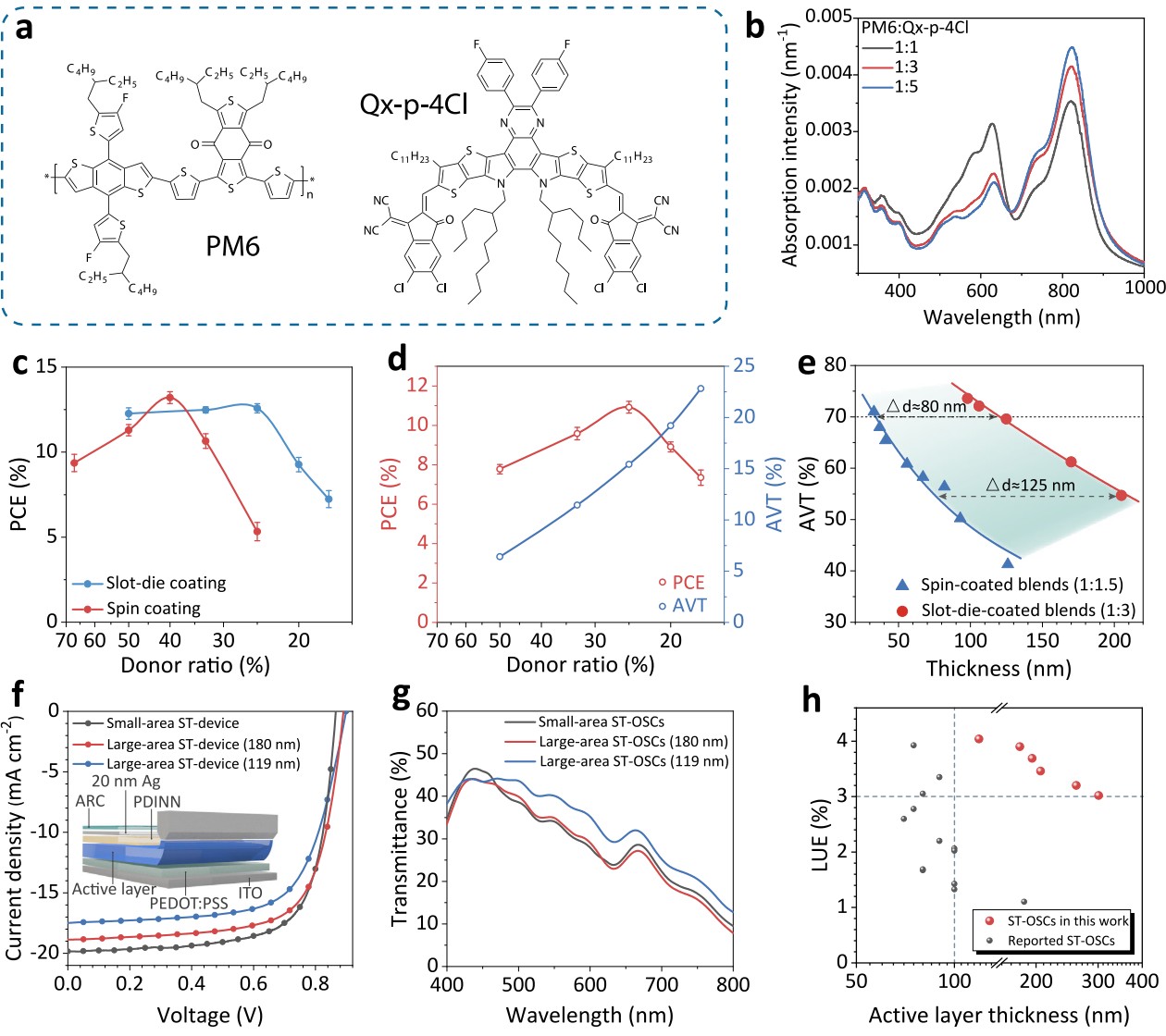

**Fig. 1 | Device characteristics. a** Chemical structures of PM6 and Qx-p-4Cl. **b**. The thickness-normalized absorption spectra of PM6:Qx-p-4Cl films. The device parameters statistics of (**c**) opaque and (**d**) semitransparent OSCs with different D:A ratios. **e** The change of AVT against active layer thickness for spin-coated and slot-die-coated PM6:Qx-p-4Cl blends (Δd represents the thickness difference between 1:1.5 and 1:3 blends at the same transmittance). **f** The J-V curves of slot-die-coated ST-devices with MoO₃ ARC, and the insert shows the corresponding device architecture. **g** The transmittance spectra of small-area and large-area ST-devices. **h** The summary of LUE versus active layer thickness of reported large-area ST-OSCs (≥1 cm²).

ST-device with a small active layer area (0.0256 cm²) yields a $V_{OC}$ of 0.87 V, $J_{SC}$ of 19.85 mA cm⁻², FF of 71.66% and PCE of 12.37%. For large-area ST-OSCs with different thicknesses, the 1 cm² device with 180 nm active layer shows the highest PCE of 11.81%, along with a $V_{OC}$ of 0.89 V, $J_{SC}$ of 18.86 mA cm⁻² and FF of 70.32%. When the active layer thickness is increased to 301 nm, the PCE is slightly reduced to 11.25%. It is worth noting that the introduction of ARC improves the AVT of thick-film ST-devices to more than 30% (Fig. 1g and Supplementary Figs. 10–12), indicating outstanding optical behaviors as shown in Supplementary Fig. 14.

The parameter of LUE is calculated to reveal the balance between PCE and AVT.[1,10,20] The LUE of 4.00% is achieved in a small-area device. Besides, Fig. 1h and Supplementary Table 1 summarize the LUE values of reported large-area ST-OSCs with different thicknesses. The high LUE above 3% is mostly achieved by decreasing the active layer thickness to below 90 nm. Increasing thickness leads to a significant drop in the transmittance of the undiluted devices (Fig. 1e), thereby reducing the LUE to around 2% or even below 1.5% for large-area ST-

OSCs with thicknesses larger than 100 nm. In contrast, the high carrier mobility (μ) of the Qx-p-4Cl system ($\mu_{hole}$ = 6.95×10⁻⁴ cm²V⁻¹ s⁻¹, $\mu_{electron}$ = 5.79×10⁻⁴ cm²V⁻¹ s⁻¹) ensures high efficiency in thick-film devices as presented in Supplementary Fig. 15. Importantly, as the thickness increases from 119 nm to 301 nm, the LUE of the PM6:Qx-p-4Cl-based donor-diluted device decreases from 4.04 to 3.02%, exhibiting remarkable thickness tolerance. As far as we know, the 1 cm² ST-devices with 119 nm thickness fabricated in this work could be listed as one of the high-performing large-area ST-OSCs in terms of high LUE.

## Film morphology and photophysics characteristics

The investigation of film morphology is crucial to understanding the device performance. First, grazing incident wide-angle X-ray scattering (GIWAXS) technology was performed to study the multiscale morphological evolution.[34–36] As shown in Fig. 2a and Supplementary Fig. 16–17, all the BHJ blends prefer face-on orientation due to the strong (010) signal along the out-of-plane (OOP) direction.[35,37,38] The parameters of crystalline coherence length (CCL) and π-π stacking

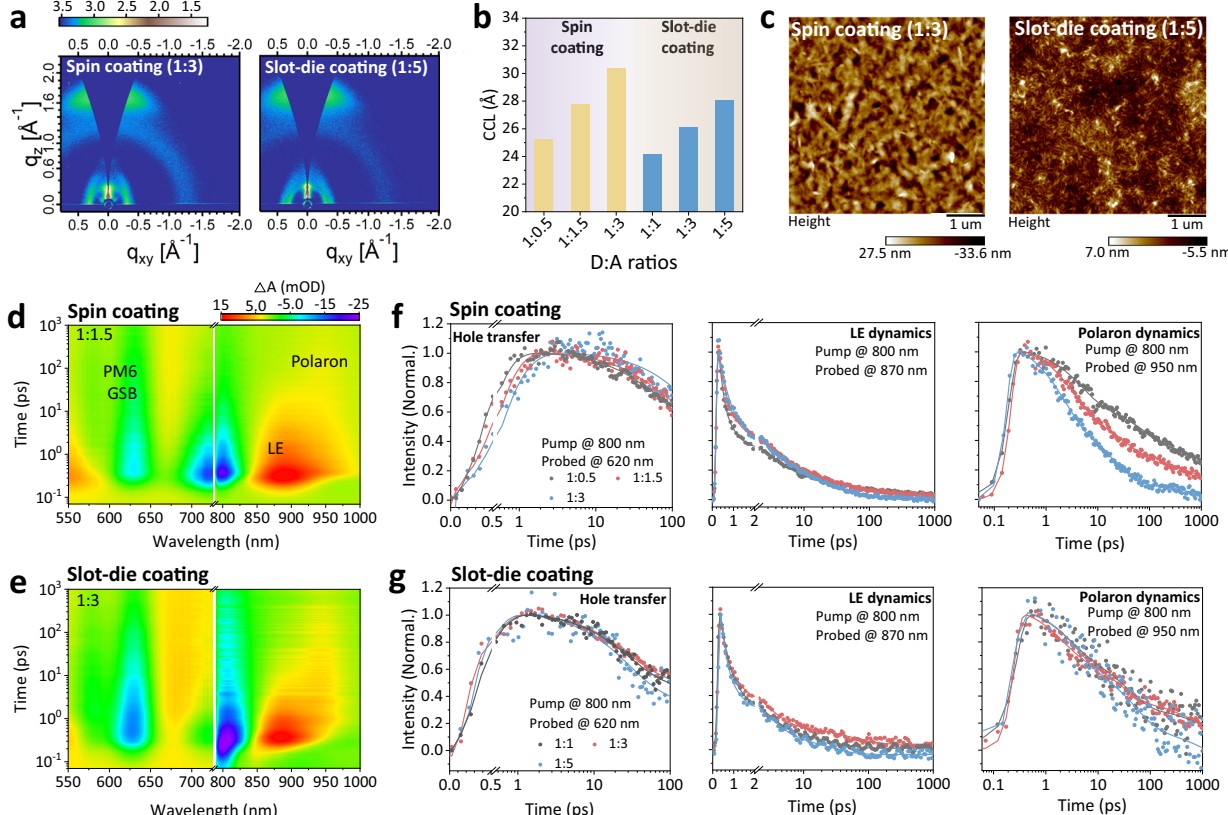

**Fig. 2 | Morphology and photophysics characteristics. a** The 2D GIWAXS patterns of PM6:Qx-p-4Cl donor-diluted films processed by spin coating and slot-die coating. **b** The summary of CCL values. **c** AFM height images of donor-diluted spin-coated and slot-die-coated films. TA maps for (**d**) spin-coated film (1:1.5) and (**e**) slot-die-coated film (1:3). The hole transfer, LE dynamics, and polaron kinetics of (**f**) spin-coated and (**g**) slot-die-coated films with different D:A ratios.

distance (d) were extracted from GIWAXS patterns to gain deeper insight into molecular packing behaviors.[36,38,39] The calculated results are presented in Fig. 2b and Supplementary Table 9. The spin-coated films exhibit an elevated CCL of the (010) π-π stacking peak in the OOP direction with increasing acceptor content. The CCL of spin-coated films with 1:0.5, 1:1.5, and 1:3 D:A ratios are 25.25, 27.81, and 30.41 Å, respectively, attributed to the excessive crystallinity in the 1:3 ratio film.[40,41] Meanwhile, a similar trend is observed in slot-die-coated films, but the CCLs of slot-die-coated films with different D:A ratios are lower than in spin coating cases overall. It can be seen that the donor-diluted slot-die-coated film has more reasonable crystallinity relative to the spin-coated film.

In addition, atomic force microscopy (AFM) was carried out to gain the surface morphological characteristics as shown in Fig. 2c and Supplementary Fig. 18-21.[42–44] The aggregation of Qx-p-4Cl within spin-coated films is enhanced with increasing acceptor ratio, accompanied by improved root mean square ($R_q$) from 3.29 nm at 1:0.5 to 8.57 nm at 1:3 as presented in Supplementary Table 10. An excessive aggregation of Qx-p-4Cl is shown in the spin-coated film with 1:3 ratio, which is consistent with the results of GIWAXS. For slot-die-coated films, all the films with different D:A ratios exhibit consistent surface morphology with lower $R_q$ than spin-coated films. Notably, the obvious fibril-like structure with fibril width in the range of 20-21 nm can be observed even at an extreme D:A ratio (1:5), and diluting donor content has a negligible effect on fibril size as shown in Supplementary Fig. 19. Combined with the GIWAXS results, the appearance of excessive crystallinity-induced aggregation in the donor-diluted spin-coated film (1:3) can lead to inferior device performance.[40,41] On the contrary, all the slot-die-coated films exhibit

more reasonable molecular packing features and fiber network structure at different D:A ratios, which ensures efficient charge transport. These results may be related to the different aggregation process of the acceptor from spin coating.

According to previous reports, the morphology has been proven to determine the behaviors of the excited state within the active layer. Thus, the transient absorption (TA) spectra of spin- and slot-die-coated films were recorded to investigate the excited state kinetics.[45–48] The 800 nm pump pulse was utilized to selectively excite the Qx-p-4Cl phase individually. As shown in Fig. 2d, e and Supplementary Fig. 23–24, the rises of excited state absorption (ESA) signals located at around 875 and 950 nm are observed, corresponding to local excitation states (LEs) and polarons, respectively. Meanwhile, the ground-state bleaching (GSB) signal among the absorption range of PM6 donor emerges together with the decay of LEs ESA signal, which is assigned to the hole transfer process.[46,48] Fig. 2f shows the kinetic decay curves for spin-coated films, revealing that increasing the acceptor content suppresses the hole transfer process, as evidenced by a progressive lengthening of the rise time of the PM6 GSB signal. As summarized in Supplementary Table 11, the half-time of rise ($t_{half-time}$) for the GSB signal increases from 0.39 ps in the 1:0.5 blend to 0.50 ps in the 1:1.5 blend and further to 0.61 ps in the 1:3 blend. The early-stage lifetime of the LEs kinetic trace is also extracted to assess the dissociation of excitons, as shown in Supplementary Table 12, and the fitted lifetime grows from 0.29 in the 1:0.5 blend to 0.58 ps in the 1:3 blend. This prolonged LE lifetime implies that excitons remain localized for a longer duration before dissociating. Besides, the increased acceptor content in the spin-coated blend can accelerate the charge decay as the polaron signal of the blend with higher acceptor content

shows more rapid decay (Fig. 2f and Supplementary Table 13).[49] For slot-die-coated films, Fig. 2g demonstrates that highly efficient charge generation is achieved across all D:A ratios. The corresponding $t_{\text{half-time}}$ values fall within a narrow range of 0.27–0.36 ps, which is comparable to that of the non-diluted films fabricated with spin coating. Similarly, the polaron signal dynamics show consistent behavior across different ratios, further confirming the robustness of charge generation and transport in slot-die-coated films. The underlying mechanism for these divergent behaviors is elucidated through AFM analysis. For spin-coated films, the extreme D:A ratio in donor-diluted film results in large phase aggregation of acceptor (Fig. 2c), leading to insufficient donor/acceptor interface, which severely hampers exciton dissociation and charge transfer. Meanwhile, the loss of the interconnected network is responsible for the limitation in charge transport for donor-diluted films. In contrast, the more ideal morphology with prominent fibrous features in slot-die-coated films with different D:A ratios ensures an efficient and nearly identical charge transfer process and charge transport, indicating that the donor dilution strategy has a negligible effect on excited state dynamics in PM6:Qx-p-4Cl slot-die-coated films.

As shown in Supplementary Fig. 19-22, and Supplementary Fig. 26, we quantified the exciton diffusion length ($L_D$) and fibril width of slot-die-coated films to verify the above conclusions.[50–55] The calculated $L_D$ values are listed in Supplementary Table 14-15.[56–59] It is evident that the fibril widths in all the slot-die-coated PM6:Qx-p-4Cl films are similar (~21 nm), and the larger $L_D$ of Qx-p-4Cl (22.34 nm) enables efficient exciton dissociation, as confirmed by a photoluminescence quenching efficiency (PLQE) above 90% (Supplementary Fig. 27). The fibril widths

of PM6:Y6 films with different D:A ratios are nearly identical (~24 nm), leading to similar exciton dynamics as shown in time-resolved photoluminescence (TRPL) measurement (Supplementary Fig. 25, Supplementary Fig. 28–29, and Supplementary Table 16). In contrast, for L8-BO- and Qx-1-based systems, a slight increase in fibril width is observed with donor dilution, indicating reduced D/A interface and suppressed exciton dissociation in donor-diluted blends. These results are consistent with performance characteristics.

Combining the above results, we believe that the outstanding performance in donor-diluted slot-die-coated devices mainly originates from the nearly-identical morphology and excited state dynamics at different D:A ratios. To figure out the internal reason, the film-forming kinetics were investigated by in-situ UV-Vis spectrum technology.[35,36,44,60,61] Fig. 3a plots the original in-situ UV-Vis spectra of spin-coated and slot-die-coated films with different D:A ratios. And the corresponding absorption line profiles are shown in Fig. 3b, c, revealing processes of film-forming and material aggregation. For spin-coated BHJ films, the entire film-forming process can be divided into two stages, which are differentiated by dashed lines.[43,60,62] Stage 1 shows a decrease in peak intensity and a slow redshift of peak location during the quick solvent removal, which represents the process of thinning of the liquid film as the solvent evaporates. In stage 2, a distinct increase in peak intensity and a rapid redshift of the peak location are observed, indicating the aggregation of Qx-p-4Cl within a supersaturated film. Subsequently, the absorption signal of Qx-p-4Cl becomes stable, indicating that the film has completed the transition from liquid to solid.

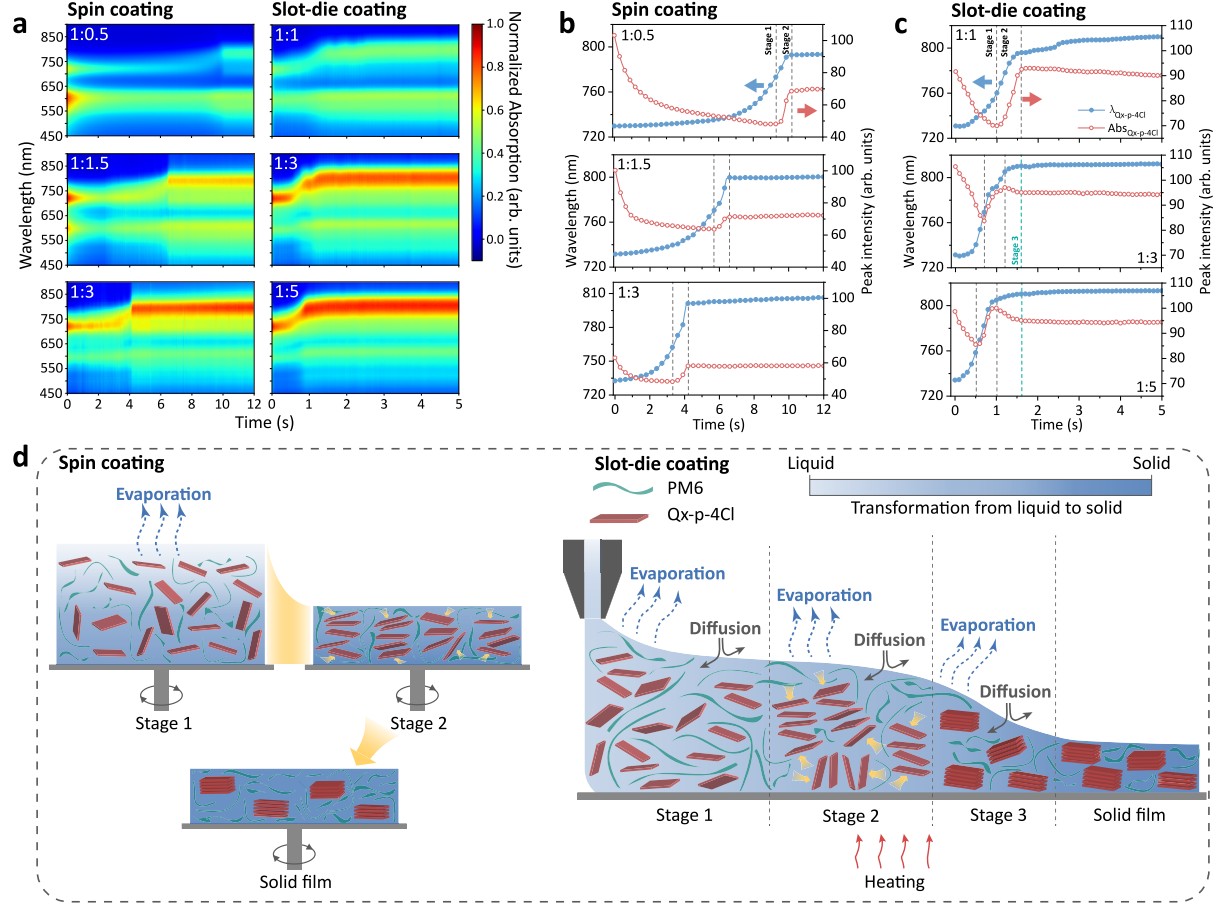

**Fig. 3 | Film formation process analysis. a** Time evolution maps of absorption spectra contour maps. Time evolution of peak location and intensity of Qx-p-4Cl in **(b)** spin-coated films and **(c)** slot-die-coated films. The blue and red arrows

represent the left vertical axis (the wavelength scale) and right vertical axis (the peak intensity scale), respectively. **d** The schematic illustration of the film-formation process of spin-coated and slot-die-coated films.

The schematic diagram in Fig. 3d clearly shows the detailed film-formation process. We focused on the duration of stages 1 and 2 to investigate the behavior of acceptor aggregation during the film-forming process. First, the spin-coated films with D:A ratios of 1:0.5, 1:1.5, and 1:3 exhibit obvious differences in stage 1, corresponding to durations of approximately 9.5, 5.7, and 3.5 s, respectively. This difference is associated with solution viscosity, which is listed in Supplementary Table 18. Diluting donor content reduces the viscosity of the BHJ solution from 5.95 mPa s at D:A ratio of 1:0.5 to 1.85 mPa s at 1:3, accelerating the process of the liquid film reaching supersaturation.[63] In stage 2, although all the spin-coated films show a similar duration of around 0.7 s, the film with higher acceptor content displays a larger redshift of the Qx-p-4Cl absorption peak. The films with 1:0.5, 1:1.5 and 1:3 ratios undergo the redshift to around 793, 800 and 806 nm, respectively, suggesting that increasing acceptor content can enhance the crystallinity-induced aggregation of acceptor during stage 2.

For slot-die-coated films, those with different D:A ratios exhibit a similar duration of the entire film-forming process of around 1.7 s, which is more rapid than that of spin-coated films due to the high temperature of the coating substrate. In the initial stage, increasing acceptor content shortens the corresponding duration despite a subtle difference. The durations of stage 1 for slot-die-coated films with 1:1, 1:3, and 1:5 ratios are 1.1, 0.7, and 0.5 s, respectively. When the film-forming process reaches stage 2, the corresponding duration times for all the slot-die-coated films are around 0.5 s, and their redshifts are almost identical, indicating a negligible influence of the D:A ratio on the aggregation of the acceptor. This is mainly because the hot substrate during slot-die coating can eliminate the excessive aggregation of the acceptor.[64] Subsequently, there is a distinct reduction in peak intensity and slow redshift of peak location after stage 2 in 1:3 and 1:5 films, which is marked as stage 3. It represents the further thinning of the active layer, followed by the formation of a solid film. Notably, during slot-die coating, the solvent evaporation is accompanied by lateral solution diffusion across the substrate, which is fundamentally different from spin coating. This lateral spreading results in continuous thinning of the liquid film. This behavior is significantly governed by viscosity. Low viscosity enhances the molecular diffusion and reduces solvent diffusion resistance, thereby prolonging the diffusion time. Conversely, high viscosity restricts diffusion dynamics.[65] As shown in Supplementary Table 18, the PM6:Qx-p-4Cl solutions with 1:1, 1:3, and 1:5 exhibit viscosity of 3.55, 1.85, and 1.44 mPa s, respectively. The reduced viscosity prolongs the solution diffusion process, which directly accounts for the emergence and extended duration of stage 3 observed in donor-diluted slot-die-coated blends. Notably, the 1:5 blend exhibits a longer stage 3 duration (0.7 s) compared to the 1:3 blend (0.4 s), attributed to its lower viscosity, which enhances lateral spreading and delays film solidification. More importantly, the emergence of stage 3 in the 1:3 and 1:5 slot-die-coated films demonstrates that acceptor aggregation occurs within the liquid film rather than in a supersaturated state. As illustrated in Fig. 3d, this liquid-phase aggregation enables the formation of well-defined fibrillar structures and continuous percolating networks.[64,66,67] However, the low viscosity may induce Marangoni flows during natural drying, influencing the film continuity and morphology of the active layer.[36] To avoid this issue, we adopted an airflow-assisted method to reduce the temperature gradient, thereby suppressing Marangoni flows and the coffee ring effect. These results comprehend why the ideal morphology and efficient charge generation can be achieved in slot-die-coated donor-diluted film.

## Optimization of slot-die-coated semitransparent modules

To highlight the superiority of high film thickness tolerance in preparing ST-modules, we further enlarged the active layer area to around 100 cm², which is formed from 23 sub-cells with an active layer area of 4.365 cm² as shown in Fig. 4a and Supplementary Fig. 32. The inverted device structure of indium tin oxide (ITO)/zinc oxide nanoparticle (ZnO NP)/active layer/$MoO_3$/Ag/$MoO_3$ is employed for module fabrication. Thermal evaporation is used to deposit the $MoO_3$ interlayer and top Ag electrode to enable precise control of film thickness.[68] Because the efficiency and transmittance of ST-OSCs are highly sensitive to film thickness.[7] To minimize the performance degradation caused by geometric loss, P1, P2, and P3 were laser scribed.[69–72] The width of the sub-cell is 4.5 mm and the length between P1 and P3 is 174 μm, thus showing a high geometric fill factor (GFF) of 96.28%. As shown in Fig. 4b and Supplementary Table 17, the 100 cm² ST-module exhibits PCE of 10.40% with $V_{OC}$ of 18.85 V, $J_{SC}$ of 0.79 mA cm² and FF of 69.85%. Supplementary Fig. 30-31 present the corresponding transmittance curve and color coordinate in the CIE-1931 chromaticity diagram. The 100 cm² ST-module shows AVT of 31.97% and color rendering index (CRI) of 85%, yielding an LUE of 3.32%. Supplementary Fig. 33 and Supplementary Table 19 summarize the LUE values of reported large-area ST-modules. It is worth noting that the LUE value of 100 cm² ST-OSCs prepared in this work demonstrates strong competitiveness compared to other reported ST modules.

Figure 4c summarizes the CTM of LUE ($CTM_{LUE}$) of ST-modules reported in recent works. It is found that the $CTM_{LUE}$ values of reported ST-modules are generally low, especially in 100 cm² modules (<60%), due to the large difference in film thickness between monolithic devices and module devices in ST-OSCs. Impressively, the 100 cm² ST-module fabricated in our work yields a $CTM_{LUE}$ of 83.00% relative to a small-area ST-device, which is nearly 1.5 times that of other reported 100 cm² ST-modules and is comparable to that of reported mini ST-modules (<20 cm²).[24] When the large-area ST-device (180 nm) is taken as the reference, the ST-module shows $CTM_{LUE}$ of 85.13%. Besides, similar results are also found in the summary of CTM of PCE ($CTM_{PCE}$). As shown in Supplementary Fig. 34 and Supplementary Tables 20–21, the 100 cm² ST-module shows a $CTM_{PCE}$ of 84.07% relative to small-area ST-OSC and 88.06% relative to the large-area ST-OSC (180 nm), respectively. It is revealed that the high thickness tolerance improves the $CTM_{PCE}$ of ST-module to a level comparable to opaque modules (84%-88%) and significantly suppresses the performance loss during upscaling for ST-OSCs. High film thickness tolerance can enable CTM approaching 90% for ST-module, eliminating the gap with opaque modules.

Considering the practical application of ST-OSCs, most reports reflect the optical characteristics of ST-OSCs, such as greenhouse, temperature control, etc.[5,24] The power-generating functionality is rarely considered. To comprehensively exhibit the potential application of ST-modules in BIPVs, we verified their feasibility from the perspectives of photovoltaic properties and optical properties as shown in Fig. 4d. For photovoltaic application, six encapsulated 100 cm² modules were connected in parallel as a whole power-generating window (around 600 cm² in total) to achieve image display and energy storage, which are common and necessary application scenarios in homes and offices. Supplementary Fig. 35 provides the block diagrams of driving a liquid crystal display (LCD) screen and charging the 18650 lithium-ion battery system through the semi-transparent power-generating window, which is integrated into a scaled-down house model (Fig. 4e). When the model is transferred to the sunny outdoor environment, we utilize the 600 cm² power-generating window to simultaneously power a 4.3" LCD and a lithium-ion battery. It is worth noting that the screen can display artistic images when the VGA video signal is connected to the LCD (Supplementary Fig. 36 and Supplementary Movie 1). On the other hand, the high acceptor content in the donor-diluted blend can absorb more near-infrared (NIR) light, thereby effectively blocking the NIR radiation and heat. The ST-module exhibits an excellent infrared radiation rejection (IRR) of 88.28%. As shown in Fig. 4f and Supplementary Fig. 37, another scale-down house model was exposed to the solar simulator to simulate the sunlight exposure. During the 30 min

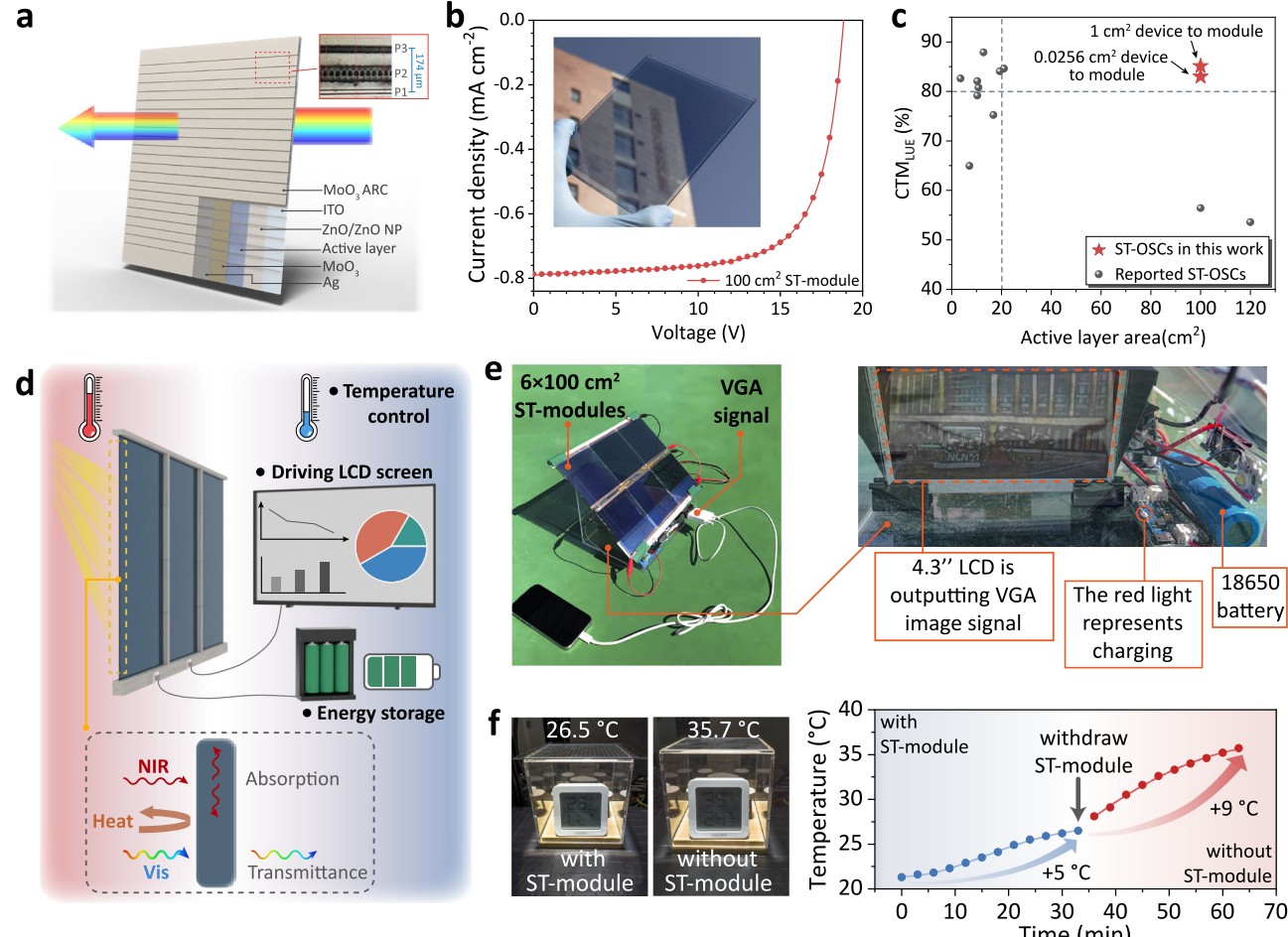

**Fig. 4 | The fabrication of ST-modules and practical applications in an outdoor environment. a** The structure of the ST-module prepared in this work. **b** The J-V curve of 100 cm² ST-module, and the insert figure shows the 100 cm² ST-module. **c** The summary of CTM$_{LUE}$ versus the active layer area of the reported large-area ST-module. **d** The schematic diagrams of practical applications verified in this work.

**e** Photos of a scale-down house model with a 600 cm² power-generating window that powers an LCD screen and charges an 18650 lithium-ion battery. **f** The relationship between the temperature inside the house model and irradiation time (the experiment is conducted with a solar simulator under AM 1.5G conditions).

continuous irradiation, the temperature inside the house model covered with ST-module "window" is increased from ambient temperature (21.4 °C) to 26.5 °C. When the ST-module is withdrawn, a significant increase in temperature from 26.5 °C to 35.7 °C is observed after 30 min of irradiation. The detailed temperature data are listed in Supplementary Table 22. This increase is approximately twice that of the house model covered with ST-module window, indicating an excellent capability of temperature control of donor-diluted ST-modules. Combined with the application verification of power-generating functionality, these results demonstrate the feasibility of ST-OSCs in BIPVs.

Device stability is also a key concern in the application of ST-OSCs. The photostability of 1 cm² donor-diluted ST-OSCs (D:A = 1:3) was tested under one sun illumination in a nitrogen atmosphere, as shown in Supplementary Fig. 38. The large-area device yields T$_{90}$ (retains ca. 90% of the initial efficiency) approaching 1000 h and extrapolated T$_{80}$ (retains ca. 80% of the initial efficiency) exceeding 3100 h, indicating excellent photostability in donor-diluted ST-OSCs. Furthermore, the outdoor stability testing for the 100 cm² ST-module was conducted to simulate real-world operating conditions better, and the weather data during the testing period were summarized in Supplementary Table 23. When the ST-module was exposed to the outdoor environment for around 1000 h, the PCE dropped to 82.61% of its initial

efficiency. It is worth noting that the extrapolated T$_{80}$ of the ST-module was calculated to 1500 h. As a result, donor-diluted ST-OSCs, including monolithic devices and modules, yield high performance and long-term stability, demonstrating excellent potential for practical applications.

## Discussion

In summary, we demonstrated that the synergy of donor-dilution strategy and slot-die coating realizes outstanding optical and device performance in ST-OSCs with a thick active layer. The optimal D:A ratio of 1:3 in 1 cm² slot-die-coated ST-OSCs based on PM6:Qx-p-4Cl enables AVT over 30% at a thickness above 200 nm and PCE of 11.81% at a thickness of 180 nm. When the thickness is reduced to 119 nm, an excellent LUE of 4.04% is achieved, along with PCE of 10.57% and AVT of 38.21%. In-depth investigations reveal that the aggregation of Qx-p-4Cl within slot-die-coated active layer occurs in the liquid film with high temperature rather than supersaturated film like spin coating, which promotes the formation of network structure and suppresses the excessive aggregation of acceptor even at the D:A ratio of 1:5. Meanwhile, the exciton diffusion length is longer than the aggregation size, ensuring the efficient charge generation in donor-diluted blends. Benefitting from these advantages, the CTM$_{LUE}$ of around 85% is obtained in 100 cm² printed ST-module, which yields LUE of 3.32%

along with PCE of 10.40% and AVT of 31.97%. Furthermore, we achieved the applications of powering LCD, energy storage and temperature control in a scale-down house model by utilizing an encapsulated power-generating window, demonstrating the feasibility of BIPVs of ST-OSCs. This work realized excellent LUE in large-area ST-devices with thick active layer and anticipated that donor-dilution strategy holds promising prospects for real-world applications of ST-OSCs.

## Methods

### Materials
The polymer PM6 and non-fullerene acceptors Qx-p-4Cl, Y6, L8-BO and Qx-1 were obtained from Hyper, Inc (China). The electron transport layer PDINN was purchased from Solarmer Materials, Inc. The solvent o-xylene was purchased from Sigma-Aldrich Inc. The additive 1-chloronaphthalene (CN) was purchased from TCI. All the materials were used in this study without any purification. The large-area glass/ITO substrates were purchased from Youxuan New Energy Technology Co., Ltd.

### Devices fabrication
The conventional device structure of ITO/PEDOT:PSS/active layer/PDINN/Ag was used in this work. The glass/ITO substrates with a sheet resistance of 15 Ω square$^{-1}$ were cleaned in deionized water, acetone, ethanol, and isopropanol with ultrasonic treatment for 30 min. Afterward, the glass/ITO substrates were treated with ultraviolet–ozone (Ultraviolet Ozone Cleaner, Jelight Company, USA) for 15 min. PEDOT:PSS as the hole transport layer was spin-cast on ITO glass at 5000 rpm for 30 s. And then, the PEDOT:PSS-coated ITO substrates were baked at 150 °C for 15 min in air. The PM6:Qx-p-4Cl (the total concentration is 20 mg mL$^{-1}$), PM6:Y6 (the total concentration is 16 mg mL$^{-1}$), PM6:L8-BO (the total concentration is 17.6 mg mL$^{-1}$) and M6:Qx1 (the total concentration is 20 mg mL$^{-1}$), with different D:A ratios were dissolved in O-xylene solvent with 0.5 vol% CN, and stirred at 80 °C overnight. For spin-coated devices, the active layer was spin-coated on PEDOT:PSS layer with 3000 rpm and 30 s. For slot-die-coated devices, the active layer was slot-die-coated on the PEDOT:PSS layer with injection speed of 160 μL min$^{-1}$, coating speed of 12 mm s$^{-1}$ and coating temperature of 90 °C for slot-die-coated devices. Next, the PDINN (1 mg mL$^{-1}$ in methanol) was spin-coated on the active layer with 3000 rpm and 30 s. Finally, the 100 nm Ag electrode was evaporated under vacuum condition with pressure of around $1.0 \times 10^{-6}$ mbar for opaque devices. For ST-OSCs, the 20 nm ultra-thin Ag electrode and 35 nm MoO$_3$ ARC were sequentially evaporated on the PDINN layer under vacuum condition with pressure of around $1.0 \times 10^{-6}$ mbar.

### Characterization
The photovoltaic characteristics of devices were measured in a N$_2$-filled glovebox. The J-V curves were measured with Newport Thermal Oriel 91159 A solar simulator under AM 1.5G condition, and the Newport Oriel PN 91150 V Si-based solar cell was used to calibrate the light intensity. And the device area for large-area devices and modules are 1 and 100 cm$^2$, respectively. Finally, the J-V curves were recorded by Keithley 2400 source-measure unit. More than 10 devices were fabricated to ensure the reproducibility of the data. Furthermore, the Oriel Newport system (Model 66902) was used to investigate the EQE of OSCs.

The space-charge-limited current (SCLC) method is adopted to calculate carrier mobility. For SCLC fitting, the Mott-Gurney law ($J = 9 \times \varepsilon_0 \mu V^2 / 8 \times L^3$) is used, where $\varepsilon_r$ is the permittivity of materials ($\varepsilon_r$ of 3 is selected in this work), $\varepsilon_0$ is the vacuum permittivity ($8.854 \times 10^{-12}$ F/m), and L is the film thickness (L of 100 nm is used in this work).

The femtosecond TA spectra were measured with the optical instrument consisting of a Ti:sapphire femtosecond laser (coherent) and an optical parametric amplifier (OPA) system. The seed pulses with

35 fs pulse width and a repetition rate of 1 kHz are generated from the amplified Ti:sapphire femtosecond laser, which can be split into two parts of laser. One for routing to the OPA to provide pump pulse, and the other for generating a broadband probe light. The energy of the pump beam was 18 nJ pulse$^{-1}$, which is too low to generate the annihilation effects of exciton–exciton and exciton–charge. GIWAXS measurement was performed by XEUSS WAXS/SAXS system, Xenocs, France. The corresponding X-ray wavelength and incident angle are 1.5418 Å and 0.18°, respectively. And 2D scattering patterns were obtained by Pilatus 300 K. AFM images were obtained from the devices directly on a VEECO Dimension 3100 atomic force microscope working under tapping mode. The film-formation process was conducted on a lab-assembled in-situ absorption test apparatus with an integrated Filmetrics F-20 spectrometer, and the absorption baseline was taken with glass/ITO/PEDOT:PSS before each coating session. For PL and TRPL measurements, the MAR165 CCD detector with time-correlated single-photo counting (TCSPC) module was used to collect the data, and the pump wavelength is 800 nm.

The L$_D$ can be calculated via fluence-dependent photoemission kinetics. The effect of annihilation on the photoemission decay is described by Eq. (1).

$$\frac{dn(t)}{dt} = -kn(t) - \gamma n^2(t) \tag{1}$$

where n(t), k and γ are the excitation density as a function of decay time $t$, and monomolecular decay constant and bimolecular recombination rate, respectively. And the γ can be obtained by fitting experimental data via Eq. (2).

$$n(t) = \frac{n(0)e^{-kt}}{1 + \frac{\gamma}{2k}n(0)[1 - e^{-kt}]} \tag{2}$$

where n(0) is the initical excitation density at $t = 0$. And L$_D$ is calculated by $L_D = \sqrt{(D\tau)}$, where $\tau$ is exciton lifetime and D is the diffusion coefficient, which is given by D=γ/(8πr) (r is the annihilation radius approximated as 2 nm).

The geometric fill factor (GFF) is defined as the ratio of the effective area of the module (A$_{effective\ area}$) (i.e., the total active layer area of all the sub-cells) to the total module area (A$_{module\ area}$):

$$GFF = \frac{A_{effective\ area}}{A_{module\ area}} = \frac{n \times l \times W}{n \times l \times (W + w)} = \frac{W}{W + w} \tag{3}$$

where n and l represent the number of sub-cells and the length of sub-cells, respectively. The W is width of the sub-cell, and the w is the the width of the dead area (i.e., the distance between P1 and P3).

The viscosity data were measured via a viscometer (Brookfield, DV2T).

### Reporting summary
Further information on research design is available in the Nature Portfolio Reporting Summary linked to this article.

## Data availability
The data generated in this study are provided in the Supplementary Information/Source Data file. Source data are provided with this paper.

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

## Acknowledgements

This work was financially supported by National Key R&D Program of China under Grant No. 2024YFA1208200 (J.Z.), the Chinese Academy of Sciences Project for Young Scientists in Basic Research under Grant No. YSBR-110 (D.D., T.W.), the National Natural Science Foundation of China under Grant Nos 52373177, 22135001 and 22309034 (J.Z., Z.W., T.W.). We thank the robotic AI-Scientist platform of Chinese Academy of Sciences for providing the organic solar module integration and fabrication system.

## Author contributions

Z.W. conceived the idea, supervised this project, and contributed to the writing of the final version. J.Z. planned the experiments, contributed to the manuscript preparation, carried out the GIWAXS test and data analysis. T.W. carried out the device and module fabrication, the characterization test and data analysis, the validation for BIPVs functionality and drafted the manuscript. J.F. provided the materials, helped to fabricate the semitransparent large-area modules and validation for BIPVs functionality, and took care of the revision of this manuscript. D.D., W.Z., and C.H. assisted in the selection of photovoltaic materials and provided the materials characteristics analysis. X.H. supported the TA test and proposed constructive opinions for the analysis of TA data. H.Z., C.T., and Y.W. assisted in the fabrication of the device and in-situ absorption test. Z.F. measured the TA spectroscopy.

## Competing interests

The authors declare no competing interests.
