## [Transparent Peer Review file · Nature Communications]

Scalable Semitransparent Organic Solar Cells with Robust Film Thickness Tolerance for Building-Integrated Photovoltaics

Corresponding Author: Professor Zhixiang Wei

Version 0:

Reviewer comments:

Reviewer #1

(Remarks to the Author)

The authors realized excellent performance in large-area translucent devices with robust film thickness tolerance, revealing the promising application prospects of ST-OSCs. It is of significant interest that the practicality of BIPVs for power generation, energy storage, and temperature regulation has been confirmed through a scaled-down model featuring a 600 cm² power-generating window. Besides, the unique aggregation mechanism and excited state dynamics in slot-die-coated donor-diluted blends were analyzed, which can support their conclusions. The manuscript is well organized and demonstrates the feasibility of ST-OSCs in BIPVs. I would like to recommend its publication in Nature Communications after minor revision.

1. In Fig. 4c, the authors achieved high CTMLUE in this work, however, the detailed values are not mentioned in the manuscript nor listed in tables. The same issue is also observed in Supplementary Fig. 34. The author should provide corresponding detailed data. Additionally, please clarify the film thickness used for the CTMLUE calculation between the 1 cm² device and the module.

2. In Fig. 3c, why did stage 3 only appear in the donor-diluted slot-die-coated blends, while the slot-die-coated blend with 1:1 ratio shows the film-formation dynamic similar to the spin-coated blends? Interestingly, stage 3 is more pronounced in the blend with lower donor content, and the duration time of stage 3 becomes longer with diluting donor content. Can authors give more explanations?

3. For AFM measurement, the authors point out that the change of D:A ratio has a subtle effect on the fiber network structure of slot-die-coated blends via detecting the surface morphology. Can D:A ratio affect the fiber size? The authors should provide some discussion on the microscopic scale.

4. The change of D:A ratio can affect the absorption characteristics of bulk heterojunction blends; however, this work lacks absorption spectroscopy measurements of PM6:Qx-p-4Cl with different D:A ratios. The authors should include the relevant results and provide further discussion.

5. Some minor mistakes should be proofed throughout the article: (1) The black data points in Fig. 1e are of different sizes; (2) The detailed temperature data in Fig. 4h should be listed in a table.

Reviewer #2

(Remarks to the Author)

In the manuscript, the authors have developed a new system, fabricated via slot-die coating with a non-halogen solvent in ambient air, that can maintain high light utilization efficiency (LUE) while showing exceptional film thickness tolerance from 100 to 300 nm. This development represents a substantial advancement for semi-transparent organic solar cells (ST-OSCs), which traditionally achieve high LUE at film thicknesses of less than 100 nm. Additionally, the authors utilized a scaled-down house model featuring a 600 cm² power-generating window to demonstrate the potential of BIPVs for power generation, energy storage, and temperature regulation, highlighting the promise of donor-diluted translucent devices for real-world

applications. I suggest that the manuscript be accepted with minor revisions.

1) Fig. 2 demonstrated both the mechanism behind the high-performance donor-diluted ST-OSCs and the generality of the donor-dilution strategy by employing a combination of TA technology, LD calculation, and multi-photovoltaic system comparative studies. In fact, there are some minor logical confusions here. Firstly, the excited state behaviors are closely related to morphology; however, this work discusses less about the influence of morphology on excited state dynamics. Secondly, the discussions in this section would benefit from phase domain size analysis, as the domain size and LD collectively affect whether exciton can reach the donor-acceptor interface. Additionally, in my opinion, the inclusion of the analysis of Eb here seems tangential to the main focus issue. Therefore, the authors need to reorganize the discussions in this section to improve the overall logic.

2) There is insufficient discussion of TA spectroscopy in this article, especially for the hole transfer process. More discussions are needed. Importantly, this manuscript would be strengthened by including a comparison of the dynamics between spin-coated and slot-die-coated blends. In addition, the information on probe wavelength and pump wavelength is missing in Fig. 2f-g, which will confuse readers. The authors need to complete the relevant information.

3) In the measurement of in-situ UV-Vis spectra, the authors proposed that the solution removal during film formation results in a drop in absorption intensity. However, the proposed explanation is inconsistent with the splashing-free nature of slot-die coating method, which distinctly differentiates it from spin coating. This manuscript should discuss alternative reasons for the absorption decrease in slot-die coating.

4) The EQE, transmittance (T) and reflectance (R) spectra are necessary for the study of ST-OSCs. This article measured the EQE and transmittance spectra, but lacked the characteristics of the reflectance spectrum. The authors need to add the reflectance spectrum and the EQE+T+R spectrum in this article to support the results of device performance.

5) The calculation method of GFF is missing. The authors need to add the information in the methods section.

6) Closely related papers should be cited, such as Nat Commun 2025, 16, 7421; Adv. Mater. 2025, 37, 2420439; Adv. Funct. Mater. 2023, 33, 2212601.

Reviewer #3

(Remarks to the Author)

In their manuscript "Scalable Semitransparent Organic Solar Cells with Robust Film Thickness Tolerance for Building-Integrated Photovoltaics", the authors present slot-die coated semitransparent organic solar cells and modules with high LUE. I find the topic relevant, but I think the following questions need to be addressed before the manuscript can be accepted in Nature Communications:

- The authors speak of translucent cells and modules. Is this used with the same meaning as semitransparent, or is there a difference?
- I do not completely understand the cell-to-module efficiency remaining ratio (CTM) that the authors introduce. Do they compare the efficiency of the semitransparent module to the opaque cell? That would not be a fair comparison. If they compare the semitransparent cell with the semitransparent module, I do not understand why the values is so low.
- How general is the difference between spin-coating and slot-die coating that is shown in Figure 1b? The authors have investigate the morphology and photophysics for different blends, have they also analyzed the device performance?
- On page 3, the authors write that thin active layers face exacerbated performance degradation when up-scaled. Can they clarify what they mean by performance degradation, and give a reference for that?
- Transmittance spectra and photographs of the cells are given in the Supporting Information, but not in the main text. I think they are relevant for the content of the manuscript and at least some of the spectra should be moved to the main manuscript. On the other hand, some of the details shown in Figure 4 could go to the Supporting Information.
- If AVT values are given (for instance on page 5), the corresponding spectrum should be referenced.
- The authors should perform a consistency check of EQE + Transmittance < 100% for at least some of the devices.
- If I understand correctly, the slot-die coated modules were fabricated with an evaporated silver top electrode and MoO₃ layer. The latter is not really convenient for large-area production. Can the authors comment on this?
- Since the authors highlight the applications of semitransparent OPV, they should include operational stability of their modules.

Reviewer #4

(Remarks to the Author)

The manuscript reports a donor-dilution strategy for fabricating thick-film slot-die-coated ST-OSCs using non-halogenated solvents under ambient conditions. The authors demonstrate devices with high efficiency, film thickness tolerance, and light utilization efficiency, supported by mechanistic insights into aggregation and charge-generation dynamics. Furthermore, the approach is scaled to 100 cm² translucent module, which exhibits color quality and the practical potential of donor-diluted devices is validated through a 600 cm² power-generating window in a model house, thereby highlighting their relevance for BIPV applications. However, there are still some questions and issues that should be addressed.

1. This manuscript lacks data on color rendering index and infrared rejection rate. Without this information, it is difficult to evaluate the semitransparent and thermal insulation performance quantitatively. Therefore, the authors should provide the following relevant data: color rendering index, infrared rejection spectra and geometrical fill factor.
2. It is imperative to acknowledge the pivotal role of high carrier mobility in the fabrication of OSCs, particularly regarding their thickness tolerance. It is recommended that the authors provide data on the electron and hole mobility of single-component and blend films. This will demonstrate the materials' potential for use in the production of module devices with thickness of approximately 300 nm.
3. The manuscript contains some inaccuracies about the use and definition of certain abbreviations. For instance, the abbreviation "CCL" should be "crystalline coherence length" instead of "coherence length". It is therefore recommended that authors undertake a systematic review of the entire manuscript to ensure that all abbreviations are accurately defined at first

mention and used consistently throughout.

4. The manuscript proposes that donor dilution lowers the viscosity of the BHJ solution, thereby accelerating the transition of the liquid film to a supersaturated state. However, the lack of viscosity measurements for BHJ solutions with varying donor content hinders the robustness of this mechanistic explanation. The authors should therefore provide corresponding viscosity data to support their claim.

5. The photoluminescence quenching efficiency of the optimal D:A ratio of 1:3 PM6:Qx-p-4Cl should be given to better demonstrate enough exciton dissociation.

6. The manuscript proposes that donor dilution reduces solution viscosity and thereby influences film formation dynamics. While low-viscosity solutions have been shown to enhance leveling, accelerated drying may induce Marangoni flows or abnormal phase separation, which could potentially compromise the continuity of the film. The suppression of Marangoni effects is widely recognized as a key factor in achieving favorable morphology. However, the absence of discussion on this aspect, as well as the lack of direct supporting evidence, weakens the mechanistic interpretation.

Version 1:

Reviewer comments:

Reviewer #2

(Remarks to the Author)

The authors have reasonably addressed the issues raised by this reviewer. I would like to recommend the publication of this manuscript. However, I would suggest the authors citing a recent paper on reporting state-of-the-art large-scale semitransparent organic solar module: "Scalable Polymer for Large-area Semitransparent Organic Photovoltaics" *Joule* 2025, DOI: 10.1016/j.joule.2025.102173.

Reviewer #3

(Remarks to the Author)

The authors have answered all my questions, and generally put a lot of effort into the revisions, so in my opinion the manuscript should be published in *Nature Communications*.

The only request that I still have is to add the absolute PCE values to the degradation curve that is now shown in Figure S38, especially since this curve only shows one variation. People often use devices with lower PCE for degradation studies, and these numbers are relevant to really assess and compare operational lifetimes.

Reviewer #4

(Remarks to the Author)

I recommend accepting the revised manuscript. Some closely related work should also be cited in references, e.g. *Adv. Mater.* 2024, 36, 2305367; *Nano Energy*, 2024, 121, 109219; *Adv. Energy Mater.* 2025, 15, 2501819, etc.

A point-by-point response to the reviewers' comments

Reviewer #1 (Remarks to the Author):

The authors realized excellent performance in large-area translucent devices with robust film thickness tolerance, revealing the promising application prospects of ST-OSCs. It is of significant interest that the practicality of BIPVs for power generation, energy storage, and temperature regulation has been confirmed through a scaled-down model featuring a 600 cm² power-generating window. Besides, the unique aggregation mechanism and excited state dynamics in slot-die-coated donor-diluted blends were analyzed, which can support their conclusions. The manuscript is well organized and demonstrates the feasibility of ST-OSCs in BIPVs. I would like to recommend its publication in Nature Communications after minor revision.

1. In Fig. 4c, the authors achieved high CTM_{LUE} in this work, however, the detailed values are not mentioned in the manuscript nor listed in tables. The same issue is also observed in Supplementary Fig. 34. The author should provide corresponding detailed data. Additionally, please clarify the film thickness used for the CTM_{LUE} calculation between the 1 cm² device and the module.

Author reply: We thank the reviewer for pointing out this issue. To address the reviewer's concern, we have added the detailed CTM data in the manuscript and the corresponding table. For the evaluation of CTM_{LUE}, the 100 cm² ST-module achieves a CTM_{LUE} value of 83% relative to a small-area (0.0256 cm²) ST-OSC. When compared to a large-area (1 cm²) semitransparent device with 180 nm film thickness serving as the reference, the ST-module exhibits a CTM_{LUE} of 85.13%. Regarding the calculation of CTM of PCE (CTM_{PCE}), the 100 cm² ST-module shows a CTM_{PCE} of 84.07% relative to the small-area ST-OSC and 88.06% relative to the large-area ST-OSC (180 nm), respectively. The related descriptions have been added in the highlighted text on pages 13-14 of the revised Manuscript file and Supplementary Table 21 of the revised Supplementary file as follows:

● **On pages 13-14 in the revised Manuscript file:**

“Impressively, the 100 cm² ST-module fabricated in our work yields a CTM_{LUE} of 83.00% relative to a small-area ST-device, which is nearly 1.5 times that of other reported 100 cm² ST-modules and is comparable to that of reported mini ST-modules (< 20 cm²).²⁴ When the large-area ST-device (180 nm) is taken as the reference, the ST-module shows CTM_{LUE} of 85.13%. Besides, similar results are also found in the summary of CTM of PCE (CTM_{PCE}). As shown in Supplementary Fig. 34 and Supplementary Table 20-21, the 100 cm² ST-module shows a CTM_{PCE} of 84.07% relative to small-area ST-OSC and 88.06% relative to the large-area ST-OSC (180 nm), respectively. It is revealed that the high thickness tolerance improves the CTM_{PCE} of ST-module to a level comparable to opaque modules (84%-88%) and significantly suppresses the performance loss during upscaling for ST-OSCs.”

● **In the revised Supplementary file:**

Supplementary Table 21. The CTM of PCE and LUE of ST-OSCs, which is calculated from the equation of $CTM_{PCE} = PCE_{\text{monolithic device}}/PCE_{\text{module}}$ and $CTM_{LUE} = LUE_{\text{monolithic device}}/LUE_{\text{module}}$, respectively.

Nu m.	Monolithic device			Module			CTM _{PCE} (%)	CTM _{LUE} (%)	Ref.
	Area (cm ²)	PCE (%)	LUE (%)	Area (cm ²)	PCE (%)	LUE (%)			
1	0.09	12.27	2.50	10.8	9.54	2.02	77.75	80.80	2
2	0.09	12.27	2.50	3.6	9.74	2.06	79.38	82.60	
3	0.04	8.30	1.90	12.8	7.5	1.67	90.36	87.89	5
4	1.4	7.9	4.05	7.14	5.25	2.63	66.46	64.94	21
5	-	12.6	3.9	21	10.2	3.3	80.95	84.62	7
6	-	12.6	3.9	100	6.7	2.2	53.17	56.41	
7	1.05	2.93	2.06	10.3	2.31	1.62	78.84	79.17	12
8	1.05	1.37	1.10	10.3	1.12	0.89	81.75	82.06	
9	0.063	13.31	6.02	16.8	11.3	4.53	84.90	75.25	22
10	-	12.78	2.68	19.3	11.28	2.26	88.26	84.02	23
11	0.04	12.07	5.04	120	6.69	2.70	55.43	53.57	24
12	0.0256	12.37	4.00	100	10.40	3.32	84.07	83.00	This work
13	1	11.81	3.90	100	10.40	3.32	88.06	85.13	

2. In Fig. 3c, why did stage 3 only appear in the donor-diluted slot-die-coated blends, while the slot-die-coated blend with 1:1 ratio shows the film-formation dynamic similar to the spin-coated blends? Interestingly, stage 3 is more pronounced in the blend with lower donor content, and the duration time of stage 3 becomes longer with diluting donor content. Can authors give more explanations?

Author reply: We thank the reviewer for pointing out this issue. During the slot-die coating process, the solvent evaporation is accompanied by lateral solution diffusion, leading to continuous thinning of the liquid active layer. This behavior is significantly governed by viscosity. According to a previous report (Sci. China Mater. 2025, 68, 2799), the low viscosity enhances the molecular diffusion of donor and acceptor species and reduces the solvent diffusion resistance, prolonging the diffusion duration, while the diffusion in the blend with high viscosity is restricted. As shown in Supplementary Table 18, diluting donor content reduces the viscosity from 3.55 mPa s (1:1) to 1.33 mPa s (1:5), thereby prolonging the duration of diffusion of the solution. This extended diffusion process provides a mechanistic explanation for the emergence of stage 3 exclusively in donor-diluted blends, not in the 1:1 blend. Furthermore, the lower viscosity enhances the prominence of stage 3 and prolongs its duration. The relevant discussions were highlighted on page 12 of the revised manuscript, and the related literature (Sci. China Mater. 2025, 68, 2799) was cited as References 65 as follows:

● **On page 12 in the revised Manuscript file:**

“It represents the further thinning of the active layer, followed by the formation of a solid film. Notably, during slot-die coating, the solvent evaporation is accompanied by lateral solution diffusion across the substrate, which is fundamentally different from spin coating. This lateral spreading results in continuous thinning of the liquid film. This behavior is significantly governed by viscosity. Low viscosity enhances the molecular diffusion and reduces solvent diffusion resistance, thereby prolonging the diffusion time. Conversely, high viscosity restricts diffusion dynamics.⁶⁵ As shown in Supplementary Table 18, the PM6:Qx-p-4Cl solutions with 1:1, 1:3, and 1:5 exhibit viscosity of 3.55, 1.85, and 1.33 mPa s, respectively. The reduced viscosity prolongs

the solution diffusion process, which directly accounts for the emergence and extended duration of stage 3 observed in donor-diluted slot-die-coated blends. Notably, the 1:5 blend exhibits a longer stage 3 duration (0.7 s) compared to the 1:3 blend (0.4 s), attributed to its lower viscosity, which enhances lateral spreading and delays film solidification. More importantly, the emergence of stage 3 in the 1:3 and 1:5 slot-die-coated films demonstrates that acceptor aggregation occurs within the liquid film rather than in a supersaturated state. As illustrated in Fig. 3d, this liquid-phase aggregation enables the formation of well-defined fibrillar structures and continuous percolating networks.”

● **On page 24 in the References Section of the revised Manuscript file:**

65. Liu, C. et al. Solution viscosity-governed phase separation and aggregation kinetics enable high-efficiency, eco-friendly slot-die coated organic solar cells. *Sci. China Mater.* **68**, 2799-2808 (2025).

3. For AFM measurement, the authors point out that the change of D:A ratio has a subtle effect on the fiber network structure of slot-die-coated blends via detecting the surface morphology. Can D:A ratio affect the fiber size? The authors should provide some discussion on the microscopic scale.

Author reply: We thank the reviewer for constructive comments on this work. To explore the influence of the D:A ratio on fibril width, we performed a more in-depth analysis of AFM data from the PM6:Qx-p-4Cl slot-die-coated blends with different D:A ratios and quantified the fibril width via performing line profile analysis on AFM, as shown in Supplementary Fig. 19 of the revised Supplementary file. The films with ratios of 1:1, 1:3, and 1:5 exhibit fibril widths of 20, 21, and 21 nm, respectively, indicating a negligible effect of donor dilution on the fibril width of slot-die-coated blends. This result is related to the special film-formation process in donor-diluted slot-die-coated blend. The detailed discussions and data have been added with highlighted text on page 7 of the revised Manuscript file and Supplementary Fig. 19 in the revised Supplementary file as follows:

● **On page 7 in the revised Manuscript file:**

“For slot-die-coated films, all the films with different D:A ratios exhibit consistent surface morphology with lower R_q than spin-coated films. Notably, the obvious fibril-like structure with fibril width in the range of 20-21 nm can be observed even at an extreme D:A ratio (1:5), and diluting donor content has a negligible effect on fibril size as shown in Supplementary Fig. 19.”

● **On pages 3 and 14 in the revised Supplementary file:**

“Supplementary Figure 19 | The fibril width of slot-die-coated PM6:Qx-p-4Cl blends with different D:A ratios measured from AFM images.”

Supplementary Figure 19. The AFM height images and statistics of fibril width of slot-die-coated PM6:Qx-p-4Cl films with D:A ratios of (a) 1:1, (b) 1:3 and (c) 1:5.

4. The change of D:A ratio can affect the absorption characteristics of bulk heterojunction blends; however, this work lacks absorption spectroscopy measurements of PM6:Qx-p-4Cl with different D:A ratios. The authors should include the relevant results and provide further discussion.

Author reply: Thanks very much for the reviewer's suggestion. The absorption spectra of PM6:Qx-p-Cl with different D:A ratios (1:1, 1:3, and 1:5) were measured, as shown in Fig. 1b, to investigate the impact of donor dilution strategy on the absorption characteristics. The blend with lower donor content exhibits significantly reduced absorption capacity in the visible range, indicating great potential of the donor dilution strategy in realizing high AVT. The absorption spectra and relevant discussions were added in Fig. 1b and pages 4-6 of the revised Manuscript file, respectively, as follows:

● **On pages 4-6 in the revised Manuscript file:**

“The chemical structures of the PM6:Qx-p-4Cl system and thickness-normalized absorption spectra of the blends with different D:A ratios are illustrated in Fig. 1a-b.”

“As seen in Fig. 1b-d and Supplementary Fig. 11, diluting donor content within the active layer significantly reduces the absorption in the visible range, realizing high AVT in donor-diluted ST-OSCs.”

Fig. 1|Device characteristics. **a** Chemical structures of PM6 and Qx-p-4Cl. **b** The thickness-normalized absorption spectra of PM6:Qx-p-4Cl films. The device parameters statistics of **(c)** opaque and **(d)** semitransparent OSCs with different D:A ratios. **e** The change of AVT against active layer thickness for spin-coated and slot-die-coated PM6:Qx-p-4Cl blends (Δd represents the thickness difference between 1:1.5 and 1:3 blends at the same transmittance). **f** The J-V curves of slot-die-coated ST-devices with MoO₃ ARC, and the insert shows the corresponding device architecture. **g** The transmittance spectra of small-area and large-area ST-devices. **h** The summary of LUE versus active layer thickness of reported large-area ST-OSCs ($\geq 1 \text{ cm}^2$).

5. Some minor mistakes should be proofed throughout the article: (1) The black data points in Fig. 1e are of different sizes; (2) The detailed temperature data in Fig. 4h should be listed in a table.

Author reply: We thank the reviewer for the thoughtful review of the manuscript. We have modified Fig. 1e in the original manuscript and summarized the temperature data

in Supplementary Table 22. The description of Supplementary Table 22 was added in highlighted text on page 14 of the revised manuscript. The corresponding corrections have been added in Fig. 1h in the revised Manuscript file and Supplementary Table 22 in the revised Supplementary file as follows:

● **In the revised Manuscript file:**

Fig. 1|Device characteristics. **a** Chemical structures of PM6 and Qx-p-4Cl. **b** The thickness-normalized absorption spectra of PM6:Qx-p-4Cl films. The device parameters statistics of **(c)** opaque and **(d)** semitransparent OSCs with different D:A ratios. **e** The change of AVT against active layer thickness for spin-coated and slot-die-coated PM6:Qx-p-4Cl blends (Δd represents the thickness difference between 1:1.5 and 1:3 blends at the same transmittance). **f** The J-V curves of slot-die-coated ST-devices with MoO₃ ARC, and the insert shows the corresponding device architecture. **g** The transmittance spectra of small-area and large-area ST-devices. **h** The summary of LUE versus active layer thickness of reported large-area ST-OSCs (≥ 1 cm²).

● **On page 14 in the revised Manuscript file:**

“The detailed temperature data are listed in Supplementary Table 22.”

● **On pages 4 and 32 in the revised Supplementary file:**

“Supplementary Table 22 | The summary of temperature inside the house model.”

Supplementary Table 22. The summary of the temperature inside the house model at different irradiation time.

Irradiation time (min)	Temperature (°C)
0	21.3
3	21.6
6	21.8
9	22.3
12	22.9
15	23.5
18	24.1
21	24.9
24	25.5
27	25.9
30	26.2
33	26.5
36	28.1
39	29.1
42	30.5
45	31.6
48	32.6
51	33.3
54	34.0
57	34.6
60	35.2
63	35.7

Reviewer #2 (Remarks to the Author):

In the manuscript, the authors have developed a new system, fabricated via slot-die coating with a non-halogen solvent in ambient air, that can maintain high light utilization efficiency (LUE) while showing exceptional film thickness tolerance from 100 to 300 nm. This development represents a substantial advancement for semi-

transparent organic solar cells (ST-OSCs), which traditionally achieve high LUE at film thicknesses of less than 100 nm. Additionally, the authors utilized a scaled-down house model featuring a 600 cm² power-generating window to demonstrate the potential of BIPVs for power generation, energy storage, and temperature regulation, highlighting the promise of donor-diluted translucent devices for real-world applications. I suggest that the manuscript be accepted with minor revisions.

1. Fig. 2 demonstrated both the mechanism behind the high-performance donor-diluted ST-OSCs and the generality of the donor-dilution strategy by employing a combination of TA technology, L_D calculation, and multi-photovoltaic system comparative studies. In fact, there are some minor logical confusions here. Firstly, the excited state behaviors are closely related to morphology; however, this work discusses less about the influence of morphology on excited state dynamics. Secondly, the discussions in this section would benefit from phase domain size analysis, as the domain size and L_D collectively affect whether exciton can reach the donor-acceptor interface. Additionally, in my opinion, the inclusion of the analysis of E_b here seems tangential to the main focus issue. Therefore, the authors need to reorganize the discussions in this section to improve the overall logic.

Author reply: Thanks very much for the reviewer's suggestion. We correlated the results of excited state dynamics with morphology analysis and presented an in-depth discussion. For spin-coated blends, diluting the donor content leads to greater acceptor-phase aggregation, fewer donor/acceptor interfaces, and disruption of the fiber network structure. Therefore, the transient absorption measurement exhibits that the charge transfer, LE dynamics, and charge transport are impeded with gradually diluting donor content, as shown in Fig. 2f, Supplementary Fig. 18, and Supplementary Table 11-13. On the contrary, the slot-die-coated films with different D:A ratios exhibit more ideal morphology compared to spin-coated films and a prominent fiber network structure, which enables efficient charge transfer and transport, as shown in Fig. 2g. To provide quantitative analysis, the fibril widths and L_D are quantified as shown in Supplementary

Fig. 19-22 of the revised Supplementary file. The fibril widths are obtained as the similar method as the relevant studies (Nat. Commun., 2024, 15, 6865; Nat. Mater., 2022, 21, 656; Small, 2025, 21, 2411698; Energy Environ. Sci., 2023, 16, 1062; Adv. Mater., 2023, 35, 2208211; Adv. Energy Mater., 2024, 2404062.). These literatures are cited as 50-55 in the References Section of the revised manuscript. PM6:Qx-p-4Cl maintains a similar morphology and fibril width (~ 21 nm) across different D:A ratios. Therefore, the LD of Qx-p-4Cl (22.34 nm) is sufficient to ensure efficient exciton dynamics even in donor-diluted films, as evidenced by TRPL (Supplementary Fig. 27). Furthermore, the fibril widths of PM6:Y6 films with different D:A ratios are nearly identical (~ 24 nm), leading to similar exciton dynamics (Supplementary Fig. 28-29 and Supplementary Table 16). In contrast, for L8-BO- and Qx-1-based systems, an increase in fibril width is observed with donor dilution. It is indicated that fewer D/A interfaces in donor-diluted blends increase the exciton lifetime. The corresponding discussions are added in the highlighted text on pages 8-10 of the revised Manuscript file.

According to the reviewer's comments, we have shifted the focus of this section to the LD and the fibril width and reorganized the logic of this section. Firstly, we have relocated the discussions on the device performance based on PM6:Y6, PM6:L8-BO, and PM6:Qx-1 to the section of Photovoltaic properties (the highlighted text on page 4 of the revised Manuscript file) and provided a more comprehensive analysis. Secondly, the sections of Abstract and Discussion have been refined to maintain the logical coherence of the manuscript due to the change in logic. Next, we slightly modified Fig. 2 in the revised Manuscript file to enhance reader comprehension. Meanwhile, we have removed the summary of R_q values in Supplementary Fig. 22 of the original Supplementary file, because the relevant data have already been provided in Supplementary Table 10.

The corresponding modifications are shown as follows:

- **On pages 8-10 in the revised Manuscript file:**
“Fig. 2f shows the kinetic decay curves for spin-coated films, revealing that increasing the acceptor content suppresses the hole transfer process, as evidenced by a progressive

lengthening of the rise time of the PM6 GSB signal. As summarized in Supplementary Table 11, the half-time of rise ($t_{\text{half-time}}$) for the GSB signal increases from 0.39 ps in the 1:0.5 blend to 0.50 ps in the 1:1.5 blend and further to 0.61 ps in the 1:3 blend. The early-stage lifetime of the LEs kinetic trace is also extracted to assess the dissociation of excitons, as shown in Supplementary Table 12, and the fitted lifetime grows from 0.29 in the 1:0.5 blend to 0.58 ps in the 1:3 blend. This prolonged LE lifetime implies that excitons remain localized for a longer duration before dissociating. Besides, the increased acceptor content in the spin-coated blend can accelerate the charge decay as the polaron signal of the blend with higher acceptor content shows more rapid decay (Fig. 2f and Supplementary Table 13).⁴⁹ For slot-die-coated films, Fig. 2g demonstrates that highly efficient charge generation is achieved across all D:A ratios. The corresponding $t_{\text{half-time}}$ values fall within a narrow range of 0.27-0.36 ps, which is comparable to that of the non-diluted films fabricated with spin coating. Similarly, the polaron signal dynamics show consistent behavior across different ratios, further confirming the robustness of charge generation and transport in slot-die-coated films. The underlying mechanism for these divergent behaviors is elucidated through AFM analysis. For spin-coated films, the extreme D:A ratio in donor-diluted film results in large phase aggregation of acceptor (Fig. 2c), leading to insufficient donor/acceptor interface, which severely hampers exciton dissociation and charge transfer. Meanwhile, the loss of the interconnected network is responsible for the limitation in charge transport for donor-diluted films. In contrast, the more ideal morphology with prominent fibrous features in slot-die-coated films with different D:A ratios ensures an efficient and nearly identical charge transfer process and charge transport, indicating that the donor dilution strategy has a negligible effect on excited state dynamics in PM6:Qx-p-4Cl slot-die-coated films.

As shown in Supplementary Fig. 19-22, and Supplementary Fig. 26, we quantified the exciton diffusion length (L_D) and fibril width of slot-die-coated films to verify the above conclusions.⁵⁰⁻⁵⁵ The calculated L_D values are listed in Supplementary Table 14-15.⁵⁶⁻⁵⁹ It is evident that the fibril widths in all the slot-die-coated PM6:Qx-p-4Cl films are

similar (~ 21 nm), and the larger L_D of Qx-p-4Cl (22.34 nm) enables efficient exciton dissociation, as confirmed by a photoluminescence quenching efficiency (PLQE) above 90% (Supplementary Fig. 27). The fibril widths of PM6:Y6 films with different D:A ratios are nearly identical (~ 24 nm), leading to similar exciton dynamics as shown in time-resolved photoluminescence (TRPL) measurement (Supplementary Fig. 25, Supplementary Fig. 28-29, and Supplementary Table 16). In contrast, for L8-BO- and Qx-1-based systems, a slight increase in fibril width is observed with donor dilution, indicating reduced D/A interface and suppressed exciton dissociation in donor-diluted blends. These results are consistent with performance characteristics.”

● **On page 4 in the revised Manuscript file:**

“Besides, we summarized the photovoltaic parameters of PM6:Y6, PM6:L8-BO, and PM6:Qx-1 with different D:A ratios in Supplementary Fig. 2-9 and Supplementary Table 3-5. All the slot-die-coated systems show high performance at a high acceptor ratio. In particular, the PM6:Y6 devices achieve the optimal efficiency when the acceptor ratio exceeds twice that of the donor. However, the best performance of PM6:L8-BO and PM6:Qx-1 devices is obtained at D:A ratios of 1:1.7 and 1:1.5, respectively. Further dilution of donor content leads to a decline in device performance.”

Fig. 2| Morphology and photophysics characteristics. **a** The 2D GIWAXS patterns of PM6:Qx-p-4Cl donor-diluted films processed by spin coating and slot-die coating. **b** The summary of CCL values. **c** AFM height images of donor-diluted spin-coated and slot-die-coated films. TA maps for **(d)** spin-coated film (1:1.5) and **(e)** slot-die-coated film (1:3). The hole transfer, LE dynamics, and polaron kinetics of **(f)** spin-coated and **(g)** slot-die-coated films with different D:A ratios.

- **Abstract Section on page 1 in the revised Manuscript file:**

“Using PM6:Qx-p-4Cl as a model system, we elucidate a unique film-formation mechanism and charge generation process, demonstrating that the fiber network and suitable aggregation size are crucial for ensuring higher performance in donor-diluted ST-OSCs.”

- **Discussion Section on page 16 in the revised Manuscript file:**

“Meanwhile, the exciton diffusion length is longer than the aggregation size, ensuring the efficient charge generation in donor-diluted blends.”

- **On page 23 in References Section of the revised Manuscript file:**

50. Chen, C. et al. Molecular interaction induced dual fibrils towards organic solar cells

with certified efficiency over 20%. *Nat. Commun.* **15**, 6865 (2024).

51. Zhu, L. et al. Single-junction organic solar cells with over 19% efficiency enabled by a refined double-fibril network morphology. *Nat. Mater.* **21**, 656-663 (2022).

52. Wang, Y. et al. High-performance g-dimer acceptor-based flexible organic solar cells optimized by temperature-dependent film formation process. *Small* **21**, e2411698 (2025).

53. Chen, H. et al. A 19% efficient and stable organic photovoltaic device enabled by a guest nonfullerene acceptor with fibril-like morphology. *Energy Environ. Sci.* **16**, 1062-1070 (2023).

54. Li, D. et al. Fibrillization of non-fullerene acceptors enables 19% efficiency pseudo-bulk heterojunction organic solar cells. *Adv. Mater.* **35**, e2208211 (2023).

55. Feng, W. et al. Rational design of two well-compatible dimeric acceptors through regulating chalcogen-substituted conjugated backbone enable ternary organic solar cells with 19.4% efficiency. *Adv. Energy Mater.* 2404062 (2024).

● **On pages 3 and 14-16 in the revised Supplementary file:**

“Supplementary Figure 19 | The fibril width of slot-die-coated PM6:Qx-p-4Cl blends with different D:A ratios measured from AFM images.

Supplementary Figure 20 | The fibril width of slot-die-coated PM6: Y6 blends with different D:A ratios measured from AFM images.

Supplementary Figure 21 | The fibril width of slot-die-coated PM6:L8-BO blends with different D:A ratios measured from AFM images.

Supplementary Figure 22 | The fibril width of slot-die-coated PM6:Qx-1 blends with different D:A ratios measured from AFM images.”

Supplementary Figure 19. The AFM height images and statistics of fibril width of slot-die-coated PM6:Qx-p-4Cl films with D:A ratios of (a) 1:1, (b) 1:3 and (c) 1:5.

Supplementary Figure 20. The AFM height images and statistics of fibril width of slot-die-coated PM6:Y6 films with different D:A ratios.

Supplementary Figure 21. The AFM height images and statistics of fibril width of slot-die-coated PM6:L8-BO films with different D:A ratios.

Supplementary Figure 22. The AFM height images and statistics of fibril width of slot-die-coated PM6:Qx-1 films with different D:A ratios.

2. There is insufficient discussion of TA spectroscopy in this article, especially for the hole transfer process. More discussions are needed. Importantly, this manuscript would be strengthened by including a comparison of the dynamics between spin-coated and slot-die-coated blends. In addition, the information on probe wavelength and pump wavelength is missing in Fig. 2f-g, which will confuse readers. The authors need to complete the relevant information.

Author reply: We thank the reviewer for pointing out this issue. Following the reviewer's comments, we have expanded the discussions on the transient absorption characteristics and compared the excited state dynamics between spin-coated and slot-die-coated films. As shown in Fig. 2f, the spin-coated films with different D:A ratios exhibit a significant difference. The values of half-time of rising of PM6 GSB signal ($t_{\text{half-time}}$), which represents the kinetics of hole transfer, of 1:0.5, 1:1.5 and 1:3 D:A ratios are 0.39, 0.50, and 0.61 ps, respectively. The consistent results are also observed in the measurement of excitation states (LEs), as shown in Supplementary Table 12. In addition, the dilution of donor content accelerates the charge decay as the polaron signal

of the spin-coated blend with higher acceptor content shows more rapid decay (Fig. 2f and Supplementary Table 13). In contrast, the nearly identical charge transfer processes and LE dynamics are observed in slot-die-coated films with different D:A ratios due to the more ideal morphology than donor-diluted spin-coated films (Fig. 2g). The corresponding $t_{\text{half-time}}$ values fall within a narrow range of 0.27-0.36 ps, which is comparable to that of the non-diluted films fabricated with spin coating. Meanwhile, the prominent fibrous feature ensures similar decay curves of the polaron signal in all the slot-die-coated films. For clarity, the information on probe wavelength and pump wavelength has been included in Fig. 2f-g, and the LE dynamics curves in the original Supplementary file are shifted to Fig. 2 in the revised manuscript. Meanwhile, we removed Fig. 2h in the original manuscript because the L_D data have already been listed in Supplementary Table 15, and Fig. 2i in the original manuscript is shifted to Supplementary Fig. 25 in the revised Supplementary file. The related discussions and revised Fig. 2 can be found on pages 8-10 of the revised Manuscript file as follows:

● **On pages 8-10 in the revised Manuscript file:**

“Fig. 2f shows the kinetic decay curves for spin-coated films, revealing that increasing the acceptor content suppresses the hole transfer process, as evidenced by a progressive lengthening of the rise time of the PM6 GSB signal. As summarized in Supplementary Table 11, the half-time of rise ($t_{\text{half-time}}$) for the GSB signal increases from 0.39 ps in the 1:0.5 blend to 0.50 ps in the 1:1.5 blend and further to 0.61 ps in the 1:3 blend. The early-stage lifetime of the LEs kinetic trace is also extracted to assess the dissociation of excitons, as shown in Supplementary Table 12, and the fitted lifetime grows from 0.29 in the 1:0.5 blend to 0.58 ps in the 1:3 blend. This prolonged LE lifetime implies that excitons remain localized for a longer duration before dissociating. Besides, the increased acceptor content in the spin-coated blend can accelerate the charge decay as the polaron signal of the blend with higher acceptor content shows more rapid decay (Fig. 2f and Supplementary Table 13).⁴⁹ For slot-die-coated films, Fig. 2g demonstrates that highly efficient charge generation is achieved across all D:A ratios. The corresponding $t_{\text{half-time}}$ values fall within a narrow range of 0.27-0.36 ps, which is

comparable to that of the non-diluted films fabricated with spin coating. Similarly, the polaron signal dynamics show consistent behavior across different ratios, further confirming the robustness of charge generation and transport in slot-die-coated films. The underlying mechanism for these divergent behaviors is elucidated through AFM analysis. For spin-coated films, the extreme D:A ratio in donor-diluted film results in large phase aggregation of acceptor (Fig. 2c), leading to insufficient donor/acceptor interface, which severely hampers exciton dissociation and charge transfer. Meanwhile, the loss of the interconnected network is responsible for the limitation in charge transport for donor-diluted films. In contrast, the more ideal morphology with prominent fibrous features in slot-die-coated films with different D:A ratios ensures an efficient and nearly identical charge transfer process and charge transport, indicating that the donor dilution strategy has a negligible effect on excited state dynamics in PM6:Qx-p-4Cl slot-die-coated films.”

Fig. 2| Morphology and photophysics characteristics. a The 2D GIWAXS patterns of PM6:Qx-p-4Cl donor-diluted films processed by spin coating and slot-die coating. **b** The summary of CCL values. **c** AFM height images of donor-diluted spin-coated and slot-die-coated films. TA maps for **(d)** spin-coated film (1:1.5) and **(e)** slot-die-coated

film (1:3). The hole transfer, LE dynamics, and polaron kinetics of (f) spin-coated and (g) slot-die-coated films with different D:A ratios.

● On pages 3 and 17 in the revised Supplementary file:

“Supplementary Figure 25 | The summary of exciton lifetimes.”

Supplementary Figure 25. The summary of exciton lifetimes normalized against 1:0.5 films.

3. In the measurement of in-situ UV-Vis spectra, the authors proposed that the solution removal during film formation results in a drop in absorption intensity. However, the proposed explanation is inconsistent with the splashing-free nature of slot-die coating method, which distinctly differentiates it from spin coating. This manuscript should discuss alternative reasons for the absorption decrease in slot-die coating.

Author reply: We thank the reviewer for pointing out this issue. In Fig. 3d, we have considered only solvent evaporation during the film-formation process, while neglecting other possible solvent behaviors. In fact, the solvent removal mechanism during the film-formation process in slot-die coating is distinct from that in spin coating. It is worth noting that the solvent evaporation is accompanied by lateral solution diffusion during slot-die coating, resulting in continuous thinning of the liquid active layer. This behavior is significantly governed by viscosity. Low viscosity enhances the molecular diffusion and reduces solvent diffusion resistance, thereby prolonging the diffusion time. Conversely, high viscosity restricts diffusion dynamics. In this work,

diluting donor content reduces the viscosity, as shown in Supplementary Table 18, thereby promoting the emergence of stage 3 in donor-diluted blends. According above results, we have revised the film-formation mechanism in Fig. 3d. The relevant discussions were added in highlight text on page 12 of the revised manuscript, and Fig. 3d was also revised as follows:

- **On page 12 in the revised Manuscript file:**

“It represents the further thinning of the active layer, followed by the formation of a solid film. Notably, during slot-die coating, the solvent evaporation is accompanied by lateral solution diffusion across the substrate, which is fundamentally different from spin coating. This lateral spreading results in continuous thinning of the liquid film. This behavior is significantly governed by viscosity. Low viscosity enhances the molecular diffusion and reduces solvent diffusion resistance, thereby prolonging the diffusion time. Conversely, high viscosity restricts diffusion dynamics.⁶⁵ As shown in Supplementary Table 18, the PM6:Qx-p-4Cl solutions with 1:1, 1:3, and 1:5 exhibit viscosity of 3.55, 1.85, and 1.33 mPa s, respectively. The reduced viscosity prolongs the solution diffusion process, which directly accounts for the emergence and extended duration of stage 3 observed in donor-diluted slot-die-coated blends. Notably, the 1:5 blend exhibits a longer stage 3 duration (0.7 s) compared to the 1:3 blend (0.4 s), attributed to its lower viscosity, which enhances lateral spreading and delays film solidification. More importantly, the emergence of stage 3 in the 1:3 and 1:5 slot-die-coated films demonstrates that acceptor aggregation occurs within the liquid film rather than in a supersaturated state. As illustrated in Fig. 3d, this liquid-phase aggregation enables the formation of well-defined fibrillar structures and continuous percolating networks.”

- **On page 11 in the revised Manuscript:**

Fig. 3|Film formation process analysis. **a** Time evolution maps of absorption spectra contour maps. Time evolution of peak location and intensity of Qx-p-4Cl in **(b)** spin-coated films and **(c)** slot-die-coated films. **d** The schematic illustration of the film-formation process of spin-coated and slot-die-coated films.

4. The EQE, transmittance (T) and reflectance (R) spectra are necessary for the study of ST-OSCs. This article measured the EQE and transmittance spectra, but lacked the characteristics of the reflectance spectrum. The authors need to add the reflectance spectrum and the EQE+T+R spectrum in this article to support the results of device performance.

Author reply: We thank the reviewer for the constructive comment. In order to enhance the comprehensiveness of the data presented in this study, we measured the reflectance and EQE+T+R spectra of three representative ST-OSCs, including a small-area semitransparent device, large-area semitransparent devices with thicknesses of 119 and 180 nm. The corresponding results have been provided in Supplementary Fig. 12, and

the transmittance spectra of large-area ST-OSCs with other thicknesses (195, 206, 260, and 301 nm) were transferred from Supplementary Fig. 12 of the original Supplementary file to Supplementary Fig. 10 of the revised Supplementary file. The related discussions and revised figures have been shown in highlighted text on page 5 of the revised manuscript and revised Supplementary file as follows:

- **On page 5 in the revised Manuscript file:**

“The corresponding EQE spectra are shown in Supplementary Fig. 10. The slot-die-coated ST-device with a small active layer area (0.0256 cm^2) yields a V_{OC} of 0.87 V, J_{SC} of 19.85 mA cm^{-2} , FF of 71.66% and PCE of 12.37%. For large-area ST-OSCs with different thicknesses, the 1 cm^2 device with 180 nm active layer shows the highest PCE of 11.81%, along with a V_{OC} of 0.89 V, J_{SC} of 18.86 mA cm^{-2} , and FF of 70.32%. When the active layer thickness is increased to 301 nm, the PCE is slightly reduced to 11.25%. It is worth noting that the introduction of ARC improves the AVT of thick-film ST-devices to more than 30% (Fig. 1f and Supplementary Fig. 10-12), indicating outstanding optical behaviors as shown in Supplementary Fig. 14.”

- **On pages 2 and 9-10 in the revised Supplementary file:**

“Supplementary Figure 10 | EQE and transmittance spectra of 1 cm^2 slot-die-coated semitransparent PM6:Qx-p-4Cl devices.”

“Supplementary Figure 12 | The EQE, reflection, and transmittance spectra of PM6:Qx-p-4Cl 1 cm^2 slot-die-coated semitransparent devices with ARC.”

Supplementary Figure 10. The EQE spectra of slot-die-coated (a) PM6:P4Cl semitransparent devices with different D:A ratios, (b) MoO₃-modified PM6:Qx-p-4Cl (1:3) semitransparent devices with different device area, and (c) larger active layer thickness. (d) The transmittance spectra of the MoO₃ ARC-modified large-area semitransparent device with different active layer thicknesses.

Supplementary Figure 12. The EQE, transmittance (T), reflection spectra (R), and EQE+T+R of MoO₃ ARC-modified (a) small-area semitransparent device, (b) 119 nm, and (c) 180 nm large-area semitransparent device.

5) The calculation method of GFF is missing. The authors need to add the information in the methods section.

Author reply: Thanks very much for the reviewer’s suggestion. The geometric fill factor (GFF) is defined as the ratio of the effective area of the module to the total module area, and a detailed calculation method of the GFF parameter has been added in the highlighted text on page 19 in the Methods Section of the revised manuscript as follows:

● **On page 19 in the Methods Section of the revised Manuscript file:**

“The geometric fill factor (GFF) is defined as the ratio of the effective area of the module ($A_{\text{effective area}}$) (i.e., the total active layer area of all the sub-cells) to the total module area ($A_{\text{module area}}$):

$$\text{GFF} = \frac{A_{\text{effective area}}}{A_{\text{module area}}} = \frac{n \times l \times W}{n \times l \times (W + w)} = \frac{W}{W + w} \quad (3)$$

where n and l represent the number of sub-cells and the length of sub-cells, respectively. The W is the width of the sub-cell, and the w is the width of the dead area (i.e., the distance between P1 and P3).”

6) Closely related papers should be cited, such as Nat Commun 2025, 16, 7421; Adv. Mater. 2025, 37, 2420439; Adv. Funct. Mater. 2023, 33, 2212601.

Author reply: We thank the reviewer for pointing out this issue. The literature recommended has been incorporated in paragraphs 1-2 of the Introduction Section, and cited as References 5, 6, and 14, respectively. Besides, a recent study in Joule (Joule 2025, 9, 102173) reports impressive parameters in 120 cm² ST-modules (PCE = 6.69%, AVT = 40.3%, LUE = 2.79%, CTM_{LUE} = 55.36%), offering detailed insights into the fabrication of ST-modules. To comprehensively illustrate the current progress of ST-modules, we have incorporated the device performance data into the statistical figure and table in our work (Fig. 4c, Supplementary Fig. 34, and Supplementary Table 21). This study is cited as Reference 69 in the revised Manuscript file and Reference 24 in the revised Supplementary file. The references in this part were reordered to improve the coherence. The related modifications were highlighted on pages 2, 13, 15, 19, 20 and 24 of the revised manuscript and pages 21, 32, and 36 in the revised Supplementary file as follows:

● **On pages 19, 20, and 24 in the References Section of the revised Manuscript file:**

5. Wang, D. et al. High-performance and eco-friendly semitransparent organic solar cells for greenhouse applications. *Joule* **5**, 945-957 (2021).

6. Yu, J. et al. Semitransparent organic photovoltaics with wide geographical adaptability as sustainable smart windows. *Nat. Commun.* **16**, 7421 (2025).

14. Xie, D. et al. A 2.20 eV bandgap polymer donor for efficient colorful semitransparent organic solar cells. *Adv. Funct. Mater.* **33**, 2212601 (2023).

69. Xie, D. et al. Scalable polymer for large-area semitransparent organic photovoltaics. *Joule* **9**, 102173 (2025).

● **On page 35 in the revised Supplementary file:**

24. Xie, D. et al. Scalable polymer for large-area semitransparent organic photovoltaics. *Joule* **9**, 102173 (2025).

Reviewer #3 (Remarks to the Author):

In their manuscript “Scalable Semitransparent Organic Solar Cells with Robust Film Thickness Tolerance for Building-Integrated Photovoltaics”, the authors present slot-die coated semitransparent organic solar cells and modules with high LUE. I find the topic relevant, but I think the following questions need to be addressed before the manuscript can be accepted in Nature Communications:

- The authors speak of translucent cells and modules. Is this used with the same meaning as semitransparent, or is there a difference?

Author reply: Thanks very much for the reviewer’s suggestion. In this work, the terms of “translucent” and “semitransparent” are used to express the same meaning. However, the presence of two different expressions can confuse readers. Therefore, we adopted a unified expression to ensure consistency, the term “translucent” has been replaced with “semitransparent”. In particular, the high-frequency phrases, “translucent device” and “translucent module”, have been revised to “semitransparent device (ST-device)” and “semitransparent module (ST-module)”, respectively. Throughout the Manuscript file

and Supplementary file, the term “translucent” appeared 23 times in the Manuscript file and 18 times in the Supplementary file. These modifications have been highlighted on pages 1-6, 12-16, and 25 of the revised Manuscript file and pages 2-4, 9-11, 27-28, and 31 of the revised Supplementary file.

- I do not completely understand the cell-to-module efficiency remaining ratio (CTM) that the authors introduce. Do they compare the efficiency of the semitransparent module to the opaque cell? That would not be a fair comparison. If they compare the semitransparent cell with the semitransparent module, I do not understand why the value is so low.

Author reply: Thanks very much for the reviewer’s suggestion. The parameter of CTM is introduced not to compare the semitransparent module with an opaque cell, but rather to evaluate the performance difference between a device and its module of the same type. For instance, the statistics of CTM in Fig. 4c of the manuscript and Supplementary Table 21 of the supplementary file show the ratio of performance (PCE and LUE) between the monolithic semitransparent device and the corresponding semitransparent module. Furthermore, for reference, the CTM_{PCE} values of the opaque module to opaque cell are summarized in Supplementary Table 20.

Secondly, the semitransparent modules with an area above 100 cm² reported in previous works show CTM lower than 54%, which is attributed to two main sources: (1) the design of the module and (2) scaling up production of the active layer. For the design of module structure, the presence of dead area and electrical contact resistance of the interconnects between subcells results in electrical loss and geometric loss, degrading the PCE in modules. Besides, the traditional small-area ST-OSCs maintain the high LUE threshold at approximately 60-80 nm active layer thickness, which is significantly thinner than that of high-performance opaque small-area devices (100-150 nm). The large active layer thickness is necessary to meet the requirements of large-scale manufacturing, but results in a reduction in transmittance and device performance. Consequently, the CTM of a semitransparent module is very low and typically lower

than that of an opaque module (~88%).

To achieve a high-performance semitransparent module, we adopted the donor dilution strategy to improve the PCE and AVT of thick-film ST-OSCs in this work. The 100 cm² donor-diluted semitransparent module based on PM6:Qx-p-4Cl yields CTM above 80%, which is comparable to that of the reported opaque module. The related discussions were added in highlighted text on page 3 of the revised manuscript as follows:

● **On page 3 in the revised Manuscript file:**

“Impressively, the CTM of ST-OSCs is generally lower than that of opaque OSCs. Especially for the ST-module with an area above 100 cm², the reported CTM of the 100 cm² ST-module is around 56%, which lags far behind the 100 cm² opaque module reported in previous works (~ 88%).^{23, 24} In addition to inherent electric and geometric losses of the module design, this performance gap primarily stems from the increased active layer thickness during scaling up production, which reflects the fundamental challenge of fabricating very thin film using scalable fabrication technologies. However, the large active layer thickness leads to enhanced recombination, poor charge extraction, and reduced transmittance, resulting in exacerbated degradation of LUE for ST-OSCs.^{25, 26} Consequently, it is necessary to optimize the AVT of the thick-film active layer and improve the film thickness tolerance for preparing high-performance upscaling ST-modules.^{27”}

• How general is the difference between spin-coating and slot-die coating that is shown in Figure 1b? The authors have investigate the morphology and photophysics for different blends, have they also analyzed the device performance?

Author reply: We thank the reviewer for pointing out this issue. To comprehensively analyze the performance differences between spin-coated and slot-die-coated devices, we have supplemented the discussion of device performance for PM6:Y6, PM6:L8-BO, and PM6:Qx-1 in the photovoltaic properties section. As shown in Supplementary Fig. 2-9 and Supplementary Table 3-5, all the slot-die-coated systems exhibit high

performance at high acceptor ratios. Especially for PM6:Y6, the peak efficiency of the slot-die-coated PM6:Y6 device is achieved when the acceptor ratio exceeds twice the donor. And the optimal D:A ratio for PM6:L8-BO and PM6:Qx-1 systems are 1:1.7 and 1:1.5, respectively. Morphology and TRPL measurements can well explain these results. As shown in Fig. 2 and Supplementary Figure 19-22, the fibril widths of slot-die-coated PM6:Qx-p-4Cl with different D:A ratios are similar, enabling efficient exciton dissociation and high performance in a donor-diluted device. This trend extends to the PM6:Y6 system. While a slight increase in fibril width with donor dilution is observed in L8-BO and Qx-1 systems, this results in longer exciton lifetimes and lower efficiency in donor-diluted blends. These results are consistent with the performance characteristics and demonstrate the crucial effect of morphology on charge generation in donor-diluted blends. The relevant discussions were added in highlighted text on page 4 of the revised manuscript as follows:

● **On page 4 in the revised Manuscript file:**

“Besides, we summarized the photovoltaic parameters of PM6:Y6, PM6:L8-BO, and PM6:Qx-1 with different D:A ratios in Supplementary Fig. 2-9 and Supplementary Table 3-5. All the slot-die-coated systems show high performance at a high acceptor ratio. In particular, the PM6:Y6 devices achieve the optimal efficiency when the acceptor ratio exceeds twice that of the donor. However, the best performance of PM6:L8-BO and PM6:Qx-1 devices is obtained at D:A ratios of 1:1.7 and 1:1.5, respectively. Further dilution of donor content leads to a decline in device performance.”

• On page 3, the authors write that thin active layers face exacerbated performance degradation when up-scaled. Can they clarify what they mean by performance degradation, and give a reference for that?

Author reply: We thank the reviewer for the constructive comment. In general, high-performance OSCs realize the optimal PCE at a thin active layer (100-150 nm) via spin coating technology. Especially for ST-OSCs, an ultrathin active layer (< 80 nm) is employed to balance the efficiency and transparency. However, the thin film can not

meet the requirement of scaling up production according to previous reports (Adv. Mater. 2019, 31, 1805089 and Nat. Mater., 2025, 24, 444). Firstly, it is difficult to fabricate thin films by most scalable fabrication technologies. Secondly, achieving a uniform film is challenging for large-area fabrication due to process instability. Therefore, a large number of point defects will exist in large-area OSCs, resulting in significant charge loss because the thin film is sensitive to the defects. Importantly, larger active layer thickness leads to increased recombination, reduced charge extraction, and thus exacerbated efficiency degradation, especially for the photovoltaic systems with weak thickness tolerability. Consequently, it is necessary to realize high AVT and PCE for ST-OSCs with a thick-film active layer for preparing a high-performance, large-area semitransparent module. The relevant discussions were added in highlighted text on page 3 of the revised manuscript, and the supporting literatures (Adv. Mater. 2019, 31, 1805089 and Nat. Mater., 2025, 24, 444) were cited as References 25 and 26 as follows:

- **On page 3 in the revised Manuscript file:**

“Impressively, the CTM of ST-OSCs is generally lower than that of opaque OSCs. Especially for the ST-module with an area above 100 cm², the reported CTM of the 100 cm² ST-module is around 56%, which lags far behind the 100 cm² opaque module reported in previous works (~ 88%).^{23, 24} In addition to inherent electric and geometric losses of the module design, this performance gap primarily stems from the increased active layer thickness during scaling up production, which reflects the fundamental challenge of fabricating very thin film using scalable fabrication technologies. However, the large active layer thickness leads to enhanced recombination, poor charge extraction, and reduced transmittance, resulting in exacerbated degradation of LUE for ST-OSCs.^{25, 26} Consequently, it is necessary to optimize the AVT of the thick-film active layer and improve the film thickness tolerance for preparing high-performance upscaling ST-modules.²⁷”

- **On page 21 in the References Section of the revised Manuscript file:**

25. Wang, G., Adil, M. A., Zhang, J. & Wei, Z. Large-area organic solar cells: material requirements, modular designs, and printing methods. *Adv. Mater.* **31**, e1805089 (2019).
26. Chen, H. et al. Organic solar cells with 20.82% efficiency and high tolerance of active layer thickness through crystallization sequence manipulation. *Nat. Mater.* **24**, 444-453 (2025).

• Transmittance spectra and photographs of the cells are given in the Supporting Information, but not in the main text. I think they are relevant for the content of the manuscript and at least some of the spectra should be moved to the main manuscript. On the other hand, some of the details shown in Figure 4 could go to the Supporting Information.

Author reply: Thanks very much for the reviewer's suggestion. Firstly, the transmittance spectra of small-area and large-area ST-devices with different active layer thicknesses have been added to Fig. 1g to enhance the completeness of the discussions on device performance. The sub-figure labels of Fig. 1 in the main text (pages 4-6 in the revised manuscript) have been updated due to the introduction of the transmittance spectra. Secondly, the data from Fig. 4e-f in the original manuscript have been relocated to Supplementary Fig. 35 to improve the figure clarity and streamline the main manuscript, and the content of Fig. 4 has been reorganized. In addition, the photograph of 100 cm² cell is added to the insert of Fig. 4b. The relevant revisions are shown as follows:

● **On pages 4-6 in the revised Manuscript file:**

“As seen in Fig. 1b-d and Supplementary Fig. 11, diluting donor content within the active layer significantly reduces the absorption in the visible range, realizing high AVT in donor-diluted ST-OSCs.”

“As shown in Fig. 1e, Supplementary Fig. 13, and Supplementary Table 8, we compared the AVT of 1:1.5 and 1:3 active layers with different thicknesses.”

“The current density (J)-voltage (V) curves and detailed parameters of ST-OSCs with

ARC layer are presented in Fig. 1f, Supplementary Fig. 1 and Supplementary Table 6.”

“It is worth noting that the introduction of ARC improves the AVT of thick-film ST-devices to more than 30% (Fig. 1g and Supplementary Fig. 10-12), indicating outstanding optical behaviors as shown in Supplementary Fig. 14.”

“Besides, Fig. 1h and Supplementary Table 1 summarize the LUE values of reported large-area ST-OSCs with different thicknesses.”

“Increasing thickness leads to a significant drop in the transmittance of the undiluted devices (Fig. 1e), thereby reducing the LUE to around 2% or even below 1.5% for large-area ST-OSCs with thicknesses larger than 100 nm.”

Fig. 1|Device characteristics. **a** Chemical structures of PM6 and Qx-p-4Cl. **b** The thickness-normalized absorption spectra of PM6:Qx-p-4Cl films. The device parameters statistics of **(c)** opaque and **(d)** semitransparent OSCs with different D:A ratios. **e** The change of AVT against active layer thickness for spin-coated and slot-die-

coated PM6:Qx-p-4Cl blends (Δd represents the thickness difference between 1:1.5 and 1:3 blends at the same transmittance). **f** The J-V curves of slot-die-coated ST-devices with MoO₃ ARC, and the insert shows the corresponding device architecture. **g** The transmittance spectra of small-area and large-area ST-devices. **h** The summary of LUE versus active layer thickness of reported large-area ST-OSCs ($\geq 1 \text{ cm}^2$).

● **On page 14 in the revised Manuscript file:**

“Supplementary Fig. 35 provides the block diagrams of driving a liquid crystal display (LCD) screen and charging the 18650 lithium-ion battery system through the semitransparent power-generating window, which is integrated into a scaled-down house model (Fig. 4e). When the model is transferred to the sunny outdoor environment, we utilize the 600 cm² power-generating window to simultaneously power a 4.3” LCD and a lithium-ion battery. It is worth noting that the screen can display artistic images when the VGA video signal is connected to the LCD (Supplementary Fig. 36 and Supplementary Video). On the other hand, the high acceptor content in the donor-diluted blend can absorb more near-infrared (NIR) light, thereby effectively blocking the NIR radiation and heat. The ST-module exhibits an excellent infrared radiation rejection (IRR) of 88.28%. As shown in Fig. 4f and Supplementary Fig. 37, another scale-down house model was exposed to the solar simulator to simulate the sunlight exposure.”

Fig. 4|The fabrication of ST-modules and practical applications in an outdoor environment. **a** The structure of the ST-module prepared in this work. **b** The J-V curves of 100 cm² ST-module, and the insert figure shows the 100 cm² ST-module. **c** The summary of CTM_{LUE} versus the active layer area of the reported large-area ST-module. **d** The schematic diagrams of practical applications verified in this work. **e** Photos of a scale-down house model with a 600 cm² power-generating window that powers an LCD screen and charges an 18650 lithium-ion battery. **f** The relationship between the temperature inside the house model and irradiation time (the experiment is conducted with a solar simulator under AM 1.5G conditions).

● **On pages 3 and 22 in revised Supplementary file:**

“Supplementary Figure 35 | The block diagrams of applications of power generation.

Supplementary Figure 36 | Photographs of applications of power generation.

Supplementary Figure 37 | Photograph of ambient temperature.”

Supplementary Figure 35. The block diagrams of (e) LCD drive system and (f) energy storage system.

- If AVT values are given (for instance, on page 5), the corresponding spectrum should be referenced.

Author reply: Thanks very much for the reviewer’s suggestion. The corresponding transmittance spectra have been referenced on page 5 of the revised manuscript as follows:

● **On page 5 in the revised Manuscript file:**

“As shown in Fig. 1e, Supplementary Fig. 13, and Supplementary Table 8, we compared the AVT of 1:1.5 and 1:3 active layers with different thicknesses.”

“It is worth noting that the introduction of ARC improves the AVT of thick-film ST-devices to more than 30% (Fig. 1g and Supplementary Fig. 10-12), indicating outstanding optical behaviors as shown in Supplementary Fig. 14.”

- The authors should perform a consistency check of $EQE + Transmittance < 100\%$ for at least some of the devices.

Author reply: We thank the reviewer for the valuable comment. In addition to EQE and transmittance (T) spectra, the reflectance (R) spectra are also an important parameter for ST-OSCs. To further enhance the completeness of data, the reflectance (R) of three representative ST-OSCs (small-area semitransparent device, large-area semitransparent devices with 119 and 180 nm thicknesses) was measured. Meanwhile, the EQE+T+R spectra of these representative systems were also presented in

Supplementary Fig. 12, and the content of Supplementary Fig. 10 has been restructured to improve clarity and consistency of this manuscript. The related discussions and revised figures have been shown in highlighted text on page 5 of the revised manuscript and revised Supplementary file as follows:

● **On page 5 in the revised Manuscript file:**

“The corresponding EQE spectra are shown in Supplementary Fig. 10. The slot-die-coated ST-device with a small active layer area (0.0256 cm^2) yields a V_{OC} of 0.87 V, J_{SC} of 19.85 mA cm^{-2} , FF of 71.66% and PCE of 12.37%. For large-area ST-OSCs with different thicknesses, the 1 cm^2 device with 180 nm active layer shows the highest PCE of 11.81%, along with a V_{OC} of 0.89 V, J_{SC} of 18.86 mA cm^{-2} , and FF of 70.32%. When the active layer thickness is increased to 301 nm, the PCE is slightly reduced to 11.25%. It is worth noting that the introduction of ARC improves the AVT of thick-film ST-devices to more than 30% (Fig. 1f and Supplementary Fig. 10-12), indicating outstanding optical behaviors as shown in Supplementary Fig. 14.”

● **In revised Supplementary file:**

Supplementary Figure 10. The EQE spectra of slot-die-coated (a) PM6:P4Cl

semitransparent devices with different D:A ratios, (b) MoO₃-modified PM6:Qx-p-4Cl (1:3) semitransparent devices with different device area, and (c) larger active layer thickness. (d) The transmittance spectra of the MoO₃ ARC-modified large-area semitransparent device with different active layer thickness.

Supplementary Figure 12. The EQE, transmittance (T), reflection spectra (R), and EQE+T+R of MoO₃ ARC-modified (a) small-area semitransparent device, (b) 119 nm, and (c) 180 nm large-area semitransparent device.

- If I understand correctly, the slot-die coated modules were fabricated with an evaporated silver top electrode and MoO₃ layer. The latter is not really convenient for large-area production. Can the authors comment on this?

Author reply: Thanks very much for the reviewer’s suggestion. The complexity of the evaporation process limits the application of MoO₃ and metal electrodes in large-area fabrication. However, the advantages of thermal evaporation, such as low defect density, high uniformity, excellent reproducibility, high yield, outstanding stability, etc., ensure reliable and efficient operation of the modules. More importantly, thermal evaporation enables the precise control of film thickness at the nanoscale, which is critical for semitransparent organic solar cells. Because the efficiency and transmittance of a semitransparent device are highly sensitive to film thickness. As a result, it is reasonable to adopt thermal evaporation for the fabrication of the MoO₃ interlayer and the top metal electrode in our semitransparent module. The corresponding discussions were highlighted on page 13 of the revised manuscript, and the related literature (Energy Environ. Sci., 2025, 18, 5552) was cited as References 68 as follows:

● **On page 13 of the revised Manuscript file:**

“The inverted device structure of indium tin oxide (ITO)/zinc oxide nanoparticle (ZnO NP)/active layer/MoO₃/Ag/MoO₃ is employed for module fabrication. Thermal evaporation is used to deposit the MoO₃ interlayer and top Ag electrode to enable precise control of film thickness.⁶⁸ Because the efficiency and transmittance of ST-OSCs are highly sensitive to film thickness.^{7”}

● **On page 24 in the References Section of the revised Manuscript file:**

68. Jin, Z. et al. Fully evaporated interfacial layers for high-performance and batch-to-batch reproducible organic solar modules. *Energy Environ. Sci.* **18**, 5552-5563 (2025).

• Since the authors highlight the applications of semitransparent OPV, they should include the operational stability of their modules.

Author reply: Thanks very much for the reviewer’s suggestion. To further demonstrate the application prospects of ST-SOCs in this work, we measured the operational stability of 1 cm² large-area ST-OSCs and 100 cm² ST-module. Firstly, we assessed the stability of 1 cm² large-area ST-OSCs with a 1:3 D:A ratio under maximum power point (MPP) as shown in Supplementary Fig. 38, exposed to one sun illumination from an LED light source. The large-area device yields a T₉₀ (retaining approximately 90% of the initial efficiency) approaching 1000 h and an extrapolated T₈₀ (retaining approximately 80% of the initial efficiency) exceeding 3100 h in the glove box, indicating excellent light stability in donor-diluted ST-OSCs. Secondly, the outdoor stability testing for the 100 cm² ST-module was conducted to simulate real-world operating conditions better. We tracked the efficiency of the ST-module over the testing period, and weather data, including weather type, ambient temperature, and relative humidity, were recorded simultaneously. As shown in Supplementary Fig. 38 and Supplementary Table 23, when the ST-module was exposed to the outdoor environment for around 1000 h, the PCE dropped to 82.61% of its initial efficiency. The extrapolated T₈₀ of the ST-module was calculated to 1500 h. This value is lower than that of 1 cm² ST-OSCs under continuous illumination, primarily because the module encapsulation

has not yet reached an ideal state. As a result, donor-diluted ST-OSCs, including monolithic devices and modules, yield high performance and long-term stability, demonstrating excellent potential for practical applications. The relevant discussions and stability data were added to page 15 of the revised Manuscript file and the revised Supplementary file, respectively.

- **On page 15 of the revised Manuscript file:**

“Device stability is also a key concern in the application of ST-OSCs. The photostability of 1 cm² donor-diluted ST-OSCs (D:A = 1:3) was tested under one sun illumination in a nitrogen atmosphere, as shown in Supplementary Fig. 38. The large-area device yields T₉₀ (retains ca. 90% of the initial efficiency) approaching 1000 h and extrapolated T₈₀ (retains ca. 80% of the initial efficiency) exceeding 3100 h, indicating excellent photostability in donor-diluted ST-OSCs. Furthermore, the outdoor stability testing for the 100 cm² ST-module was conducted to simulate real-world operating conditions better, and the weather data during the testing period were summarized in Supplementary Table 23. When the ST-module was exposed to the outdoor environment for around 1000 h, the PCE dropped to 82.61% of its initial efficiency. It is worth noting that the extrapolated T₈₀ of the ST-module was calculated to 1500 h. As a result, donor-diluted ST-OSCs, including monolithic devices and modules, yield high performance and long-term stability, demonstrating excellent potential for practical applications.”

- **On pages 3, 4, 24, and 33 in the revised Supplementary file:**

“Supplementary Figure 38 | Operational stability testing.”

“Supplementary Table 23 | The summary of weather data during the stability testing period.”

Supplementary Figure 38. (a) Normalized PCE with testing time for 1 cm² ST-OSCs (D:A = 1:3) and 100 cm² ST-module. (b) Normalized V_{OC}, J_{SC}, and FF with light-soaking time for 1 cm² ST-OSCs (D:A = 1:3).

Supplementary Table 23. The summary of weather data during the outdoor stability testing period.

Date	Weather	Temperature (°C)	Humidity (%)
2025/10/28-2025/11/3	Cloudy turning into clear skies	1-18	25-60
2025/11/4-2025/11/10	Light rain turning into clear skies	0-14	20-88
2025/11/11-2025/11/17	Clear skies	-4-17	17-39
2025/11/18-2025/11/24	Clear skies	-4-15	20-28
2025/11/25-2025/12/1	Clear skies	-6-9	17-37
2025/12/2-2025/12/9	Clear skies	-8-9	18-40

Reviewer #4 (Remarks to the Author):

The manuscript reports a donor-dilution strategy for fabricating thick-film slot-die-coated ST-OSCs using non-halogenated solvents under ambient conditions. The authors demonstrate devices with high efficiency, film thickness tolerance, and light utilization

efficiency, supported by mechanistic insights into aggregation and charge-generation dynamics. Furthermore, the approach is scaled to 100 cm² translucent module, which exhibits color quality and the practical potential of donor-diluted devices is validated through a 600 cm² power-generating window in a model house, thereby highlighting their relevance for BIPV applications. However, there are still some questions and issues that should be addressed.

1. This manuscript lacks data on color rendering index and infrared rejection rate. Without this information, it is difficult to evaluate the semitransparent and thermal insulation performance quantitatively. Therefore, the authors should provide the following relevant data: color rendering index, infrared rejection spectra and geometrical fill factor.

Author reply: We thank the reviewer for pointing out this issue. Firstly, the transmittance spectrum of PM6:Qx-p-4Cl (D:A = 1:3) in the near-infrared range (830-2500 nm) was measured to obtain infrared radiation rejection (IRR), as shown in Supplementary Fig. 30b. The IRR is 88.28%. Secondly, the parameters of the color rendering index (CRI) and geometrical fill factor (GFF) of the ST-module were also calculated. The ST-module exhibits CRI and GFF of 85% and 96.28%, respectively. The transmittance spectrum in the near-infrared range and relevant discussions were added in Supplementary Fig. 30b of the revised Supplementary file and pages 13-14 of the revised manuscript, respectively.

● **On pages 13-14 in the revised Manuscript file:**

“The width of the sub-cell is 4.5 mm, and the length between P1 and P3 is 174 μm, thus showing a high geometric fill factor (GFF) of 96.28%.”

“The 100 cm² ST-module shows AVT of 31.97% and color rendering index (CRI) of 85%, yielding an LUE of 3.32%.”

“The ST-module exhibits an excellent infrared radiation rejection (IRR) of 88.28%.”

● **In revised Supplementary file:**

Supplementary Figure 30. (a) The transmittance spectrum of 100 cm² ST-modules in the visible range (400-800 nm). (b) The transmittance spectrum of PM6:Qx-p-4Cl (D:A = 1:3) in the near-infrared range (830-2500 nm) and the calculation of infrared radiation rejection (IRR).

2. It is imperative to acknowledge the pivotal role of high carrier mobility in the fabrication of OSCs, particularly regarding their thickness tolerance. It is recommended that the authors provide data on the electron and hole mobility of single-component and blend films. This will demonstrate the materials' potential for use in the production of module devices with a thickness of approximately 300 nm.

Author reply: Thanks very much for the reviewer's suggestion. We measured the J-V curves of hole- and electron-only devices and obtained the hole and electron mobilities (μ_{hole} and μ_{electron}) via the space-charge-limited current (SCLC) method. As shown in Supplementary Fig. 15b, the high μ_{electron} of neat Qx-p-4Cl ($5.54 \times 10^{-4} \text{ cm}^2 \text{ V}^{-1} \text{ s}^{-1}$) is observed. And the μ_{hole} and μ_{electron} of PM6:Qx-p-4Cl (1:3) are 6.95×10^{-4} and $5.79 \times 10^{-4} \text{ cm}^2 \text{ V}^{-1} \text{ s}^{-1}$, respectively. The high carrier mobility ensures the high efficiency in thick-film devices, as shown in Supplementary Fig. 15a. In addition, the detailed calculation method is added with highlight text on page 18 in the Methods Section of the revised manuscript, and the relevant discussions are highlighted on page 6 of the revised manuscript as follows:

● **On page 6 in the revised Manuscript file:**

“In contrast, the high carrier mobility (μ) of the P4Cl system ($\mu_{\text{hole}} = 6.95 \times 10^{-4} \text{ cm}^2 \text{ V}^{-1} \text{ s}^{-1}$, $\mu_{\text{electron}} = 5.79 \times 10^{-4} \text{ cm}^2 \text{ V}^{-1} \text{ s}^{-1}$) ensures high efficiency in thick-film devices as presented in Supplementary Fig. 15. Importantly, as the thickness increases from 119 nm to 301 nm, the LUE of the PM6:Qx-p-4Cl-based donor-diluted device decreases from 4.04% to 3.02%, exhibiting remarkable thickness tolerance.”

● **On page 18 in the Methods Section of the revised Manuscript file:**

“The space-charge-limited current (SCLC) method is adopted to calculate carrier mobility. For SCLC fitting, the Mott-Gurney law ($J = 9 \times \epsilon_0 \epsilon_r \mu V^2 / 8 \times L^3$) is used, where ϵ_r is the permittivity of materials (ϵ_r of 3 is selected in this work), ϵ_0 is the vacuum permittivity ($8.854 \times 10^{-12} \text{ F/m}$), and L is the film thickness (L of 100 nm is used in this work).”

● **On pages 2 and 11 in the revised Supplementary file:**

“Supplementary Figure 15 | The PCE and LUE of ST-OSCs with different active layer thicknesses, and mobility calculation.”

Supplementary Figure 15. (a) The PCE and LUE of large-area ST-OSCs with different thicknesses. (b) The J-V curves of electron-only device based on neat Qx-p-4Cl, hole- and electron-only devices based on PM6:Qx-p-4Cl (1:3).

3. The manuscript contains some inaccuracies about the use and definition of certain

abbreviations. For instance, the abbreviation “CCL” should be “crystalline coherence length” instead of “coherence length”. It is therefore recommended that authors undertake a systematic review of the entire manuscript to ensure that all abbreviations are accurately defined at first mention and used consistently throughout.

Author reply: Thanks very much for the reviewer’s suggestion. We have updated “coherence length” to “crystalline coherence length” and conducted a systematic review of all abbreviations in this manuscript. It was found that the definitions of the abbreviations of “ITO” and “LCD” were missing in the manuscript. The definitions of ITO and LCD are indium tin oxide and liquid crystal display, respectively. The relevant updates were highlighted on pages 7, 13, and 14 of the revised manuscript as follows:

- **On page 7 in the revised Manuscript file:**

“The parameters of crystalline coherence length (CCL) and π - π stacking distance (d) were extracted from GIWAXS patterns to gain deeper insight into molecular packing behaviors.”

- **On page 13 in the revised Manuscript file:**

“The inverted device structure of indium tin oxide (ITO)/zinc oxide nanoparticle (ZnO NP)/active layer/MoO₃/Ag/MoO₃ is employed for module fabrication.”

- **On page 14 in the revised Manuscript file:**

“Supplementary Fig. 35 provides the block diagrams of driving a liquid crystal display (LCD) screen and charging the 18650 lithium-ion battery system through the semitransparent power-generating window, which is integrated into a scaled-down house model (Fig. 4e).”

4. The manuscript proposes that donor dilution lowers the viscosity of the BHJ solution, thereby accelerating the transition of the liquid film to a supersaturated state. However, the lack of viscosity measurements for BHJ solutions with varying donor content hinders the robustness of this mechanistic explanation. The authors should therefore provide corresponding viscosity data to support their claim.

Author reply: We thank the reviewer for the constructive comment on this work. To further strengthen our mechanistic interpretation, we measured the detailed viscosity of active layer solutions with different D:A ratios via a viscometer (Brookfield, DV2T), as shown in Supplementary Table 18. The viscosity values of 1:0.5, 1:1, 1:1.5, 1:3, and 1:5 are 5.95, 3.55, 2.91, 1.85, and 1.44 mPa s, respectively. It is indicated that diluting the donor content significantly reduces the viscosity of the active layer solution, which is consistent with our analysis results. The detailed viscosity data were listed in Supplementary Table 18 of the revised Supplementary file. The relevant discussions and test information on viscosity were highlighted on pages 11, 12, and 19 of the revised manuscript as follows:

● **On pages 4 and 30 in the revised Supplementary file:**

“Supplementary Table 18 | The viscosity of PM6:Qx-p-4Cl with different D:A ratios.”

Supplementary Table 18. The summary of the viscosity of PM6:Qx-p-4Cl with different D:A ratios.

D:A ratios	Viscosity (mPa s)
1:0.5	5.95
1:1	3.55
1:1.5	2.91
1:3	1.85
1:5	1.44

● **On page 11 in the revised Manuscript file:**

“This difference is associated with solution viscosity, which is listed in Supplementary Table 18. Diluting donor content reduces the viscosity of the BHJ solution from 5.95 mPa s at D:A ratio of 1:0.5 to 1.85 mPa s at 1:3, accelerating the process of the liquid film reaching supersaturation.”

● **On page 12 in the revised Manuscript file:**

“As shown in Supplementary Table 18, the PM6:Qx-p-4Cl solutions with 1:1, 1:3, and

1:5 exhibit viscosity of 3.55, 1.85, and 1.33 mPa s, respectively.”

● **On page 19 in the revised Manuscript file:**

“The viscosity data were measured via a viscometer (Brookfield, DV2T).”

5. The photoluminescence quenching efficiency of the optimal D:A ratio of 1:3 PM6:Qx-p-4Cl should be given to better demonstrate enough exciton dissociation.

Author reply: Thanks very much for the reviewer’s suggestion. We measured the photoluminescence (PL) spectra of neat Qx-p-4Cl and PM6:Qx-p-4Cl (1:3) blend to calculate the corresponding photoluminescence quenching efficiency (PLQE). The calculated PLQE of PM6:Qx-p-4Cl (1:3) blend is 90.75%, indicating efficient exciton dissociation. The PL spectra and relevant discussions were shown in Supplementary Fig. 27a of the revised Supplementary file and highlighted text on page 9 of the revised manuscript, respectively.

● **On page 9 in the revised Manuscript file:**

“It is evident that the fibril widths in all the slot-die-coated PM6:Qx-p-4Cl films are similar (~ 21 nm), and the larger L_D of Qx-p-4Cl (22.34 nm) enables efficient exciton dissociation, as confirmed by a photoluminescence quenching efficiency (PLQE) above 90% (Supplementary Fig. 27).”

● **In revised Supplementary file:**

Supplementary Figure 27. (a) The PL spectra of neat Qx-p-4Cl and PM6:Qx-p-4Cl (1:3) blend for calculation of PL quenching efficiency (PLQE). The (b) PL and (c) TRPL spectra of PM6:Qx-p-4Cl blends with different D:A ratios.

6. The manuscript proposes that donor dilution reduces solution viscosity and thereby influences film formation dynamics. While low-viscosity solutions have been shown to enhance leveling, accelerated drying may induce Marangoni flows or abnormal phase separation, which could potentially compromise the continuity of the film. The suppression of Marangoni effects is widely recognized as a key factor in achieving favorable morphology. However, the absence of discussion on this aspect, as well as the lack of direct supporting evidence, weakens the mechanistic interpretation.

Author reply: We thank the reviewer for pointing out this issue. Marangoni flows arise from surface tension gradients and play a critical role in film uniformity. During the natural drying of slot-die-coated film, the low viscosity may induce Marangoni flow. According to our previous report (Solar RRL, 2023, 7, 2300349), employing the airflow-assisted method during the slot-die coating process can reduce the temperature gradient, weakening the surface tension gradient due to the temperature dependence of surface tension. Therefore, the negative effects caused by Marangoni flows and the coffee ring effect can be suppressed. In this work, we also adopted an airflow-assisted method to weaken the Marangoni effect, realizing an ideal morphology in donor-diluted active layer as shown in Fig. 2c. The relevant discussions were added in highlight text on page 12 of the revised manuscript as follows:

● **On page 12 in the revised Manuscript file:**

“However, the low viscosity may induce Marangoni flows during natural drying, influencing the film continuity and morphology of the active layer.³⁶ To avoid this issue, we adopted an airflow-assisted method to reduce the temperature gradient, thereby suppressing Marangoni flows and the coffee ring effect.”

A point-by-point response to the remaining comments from reviewers

Reviewer #2:

The authors have reasonably addressed the issues raised by this reviewer. I would like to recommend the publication of this manuscript. However, I would suggest the authors citing a recent paper on reporting state-of-the-art large-scale semitransparent organic solar module: “Scalable Polymer for Large-area Semitransparent Organic Photovoltaics” *Joule* 2025, DOI: 10.1016/j.joule.2025.102173.

Author reply: Thanks very much for the reviewer’s suggestion. The literature recommended has been incorporated on page 14 of the manuscript, and cited as Reference 69.

- **On page 14 in the Manuscript file:**

To minimize the performance degradation caused by geometric loss, P1, P2, and P3 were laser scribed.⁶⁹⁻⁷²

- **On page 26 in the Manuscript file:**

69. Xie, D. et al. Scalable polymer for large-area semitransparent organic photovoltaics. *Joule* **9**, 102173 (2025).

Reviewer #3:

The authors have answered all my questions, and generally put a lot of effort into the revisions, so in my opinion the manuscript should be published in *Nature Communications*.

The only request that I still have is to add the absolute PCE values to the degradation curve that is now shown in Figure S38, especially since this curve only shows one variation. People often use devices with lower PCE for degradation studies, and these numbers are relevant to really assess and compare operational lifetimes.

Author reply: We thank the reviewer for pointing out this issue. The stability data including absolute PCE values have been added in Supplementary Fig. 38.

● **In Supplementary Information file:**

Supplementary Figure 38. (a) Normalized PCE with testing time for 1 cm² ST-OSCs (D:A = 1:3) and 100 cm² ST-module. (b) Normalized V_{oc}, J_{sc}, and FF with light-soaking time for 1 cm² ST-OSCs (D:A = 1:3).

Reviewer #4:

I recommend accepting the revised manuscript. Some closely related work should also be cited in references, e.g. Adv. Mater. 2024, 36, 2305367; Nano Energy, 2024, 121, 109219; Adv. Energy Mater. 2025, 15, 2501819, etc.

Author reply: Thanks very much for the reviewer's suggestion. The recommended works have been incorporated on page 14 of the manuscript, and cited as References 70, 71, and 72, respectively.

● **On page 14 in the Manuscript file:**

To minimize the performance degradation caused by geometric loss, P1, P2, and P3 were laser scribed.⁶⁹⁻⁷²

● **On page 26 in the Manuscript file:**

70. Sharma, A. et al. Semitransparent organic photovoltaics utilizing intrinsic charge generation in non-fullerene acceptors. *Adv. Mater.* **36**, e2305367 (2024).

71. Albab, M. F. et al. High-performance semi-transparent organic solar cells driven by the dipole-controlled optoelectrical response of bilateral self-assembled monolayer strategy. *Nano Energy* **121**, (2024).

72. Xie, J. et al. Non-halogenated solvent-processed organic solar cells with efficiencies exceeding 20.0% and 110 cm² modules exceeding 13% enabled by film-forming dynamics engineering. *Adv. Energy Mater.* **15**, (2025).